# Silica accelerates the selective hydrogenation of $CO_2$ to methanol on cobalt catalysts

Lingxiang Wang [1,2,7], Erjia Guan[3,7], Yeqing Wang[2,7], Liang Wang[1,4✉], Zhongmiao Gong[5], Yi Cui [5], Xiangju Meng[2], Bruce C. Gates[3] & Feng-Shou Xiao[1,2,6✉]

The reaction pathways on supported catalysts can be tuned by optimizing the catalyst structures, which helps the development of efficient catalysts. Such design is particularly desired for $CO_2$ hydrogenation, which is characterized by complex pathways and multiple products. Here, we report an investigation of supported cobalt, which is known for its hydrocarbon production and ability to turn into a selective catalyst for methanol synthesis in $CO_2$ hydrogenation which exhibits good activity and stability. The crucial technique is to use the silica, acting as a support and ligand, to modify the cobalt species via $Co-O-SiO_n$ linkages, which favor the reactivity of spectroscopically identified $^*CH_3O$ intermediates, that more readily undergo hydrogenation to methanol than the $C-O$ dissociation associated with hydrocarbon formation. Cobalt catalysts in this class offer appealing opportunities for optimizing selectivity in $CO_2$ hydrogenation and producing high-grade methanol. By identifying this function of silica, we provide support for rationally controlling these reaction pathways.

[1] Key Lab of Biomass Chemical Engineering of Ministry of Education, College of Chemical and Biological Engineering, Zhejiang University, 310027 Hangzhou, China. [2] Key Laboratory of Applied Chemistry of Zhejiang Province, Department of Chemistry, Zhejiang University, 310028 Hangzhou, China. [3] Department of Chemical Engineering, University of California, Davis, CA 95616, United States. [4] Zhejiang Provincial Key Laboratory of Advanced Chemical Engineering Manufacture Technology, Zhejiang University, 310027 Hangzhou, China. [5] Vacuum Interconnected Nanotech Workstation (Nano-X), Suzhou Institute of Nano-Tech and Nano-Bionics (SINANO), Chinese Academy of Sciences (CAS), 215123 Suzhou, China. [6] Beijing Advanced Innovation Center for Soft Matter Science and Engineering, Beijing University of Chemical Technology, 100029 Beijing, China. [7]These authors contributed equally: Lingxiang Wang, Erjia Guan, Yeqing Wang. ✉email: liangwang@zju.edu.cn; fsxiao@zju.edu.cn

The increasing atmospheric $CO_2$ concentration originating from anthropogenic emissions has caused global warming and related climate issues. Progress to reduce fossil fuel consumption and reduce $CO_2$ emissions is substantial but insufficient, and research is underway to develop processes for large-scale $CO_2$ sequestration, but validated technology is still lacking. Additional prospects for $CO_2$ reduction on a significant scale include processes for conversion of $CO_2$ as a feedstock for manufacture of platform chemicals and fuels, including CO[1,2], olefins[3,4], alcohols[5–8], and hydrocarbon fuels[9–11]. The most promising candidate routes are catalytic, including the hydrogenation of $CO_2$ to produce methanol, a large-scale platform chemical for the production of olefins, gasoline, aromatics[12], and other chemicals[13]. Further, methanol is a fuel in its own right and also promising for the storage of hydrogen[14], with the prospect of playing a significant role in hydrogen fuel cells[15]. The $CO_2$-to-methanol transformation is challenging because of the chemical inertness of $CO_2$ and the difficulty of converting it selectively to desired products.

Catalysts for the hydrogenation of $CO_2$ to methanol include supported Au particles[16], $In_2O_3$[17], Ni-Ga[18], Pd-Ga[19], Zn-Zr[20], and Mn-Co[21]. Copper, which has the advantage of being earth-abundant, has been widely investigated and applied[22–28]. Numerous copper catalysts have been designed recently to optimize interfaces between copper and metal oxide supports, because copper alone is less effective in bonding and activating $CO_2$; successful examples include $Cu/ZnO/Al_2O_3$ (used industrially for hydrogenation of CO and $CO_2$)[22–25], $Cu/ZrO_2$[26], $Cu/CeO_2$[27], and $Cu/TiO_2$[28]. In these cases, a wide scope of reaction intermediates and pathways have been identified by using the supported copper as models[22–24,26,28], but how to optimize the catalyst structure for turning the selectivity is still in need of investigation. In addition, supported copper usually suffers from deactivation caused by nanoparticle sintering under harsh reaction conditions[29,30].

Consequently, researchers have been motivated to find replacements for supported copper catalysts, focusing on inexpensive and earth-abundant metals that work effectively, such as cobalt. Cobalt is widely used in industry as a catalyst for Fischer-Tropsch synthesis[31], also drawing attention for CO oxidation[32] and ammonia synthesis[33]. But cobalt is regarded as inappropriate for the selective methanol formation, because of the its high activity for C-O dissociation[34], and CO and hydrocarbons usually form rather than methanol[35]. On the other hand, recent efforts on selectivity optimization in $CO_2$ hydrogenation have focused on engineering metal oxide supports with redox properties and electronic metal-support interactions[1,2,36–38], but the promotion role of inert supports, such as silica, has been largely overlooked.

Herein, we report how cobalt can be optimized to give efficient catalysts for methanol production by choice of a silica support. The catalyst is synthesized by incorporating cobalt nanoparticles onto amorphous silica ($Co@Si_x$) to construct abundant $Co-O-SiO_n$ interfaces, which stabilize methoxy (*$CH_3O$) species as intermediates in $CO_2$ hydrogenation. Optimizing the cobalt-to-silica ratio gives superior catalysts, even outperforming those expensive noble-metal catalysts[19] as well as the supported copper catalysts[25,28] employed for hydrogenating $CO_2$ to methanol.

## Results

**Synthesis**. The method for synthesizing $Co@Si_x$ is summarized in Fig. 1. To construct the $Co-O-SiO_n$ linkage, the hydrolysis of tetraethoxysilane (TEOS) was performed in a basic liquor containing $Co(NO_3)_2$, followed by calcination of the resultant solid at 500 °C to form a product containing predominantly $Co_3O_4$, as shown by X-ray diffraction (XRD) crystallography ($Co_3O_4@Si_x$,

Supplementary Fig. 1). The final product was obtained by reduction with hydrogen at 600 °C. The composition was adjusted by changing the amount of TEOS in the starting solution, giving $Co@Si_x$, where $x$ is the molar ratio of silica to cobalt (Supplementary Tables 1 and 2). For comparison, a conventional catalyst consisting of cobalt nanoparticles supported on silica ($Co/SiO_2$) was synthesized by a deposition method (details in the SI), the cobalt loading was 43 $wt$%.

**Catalysis in $CO_2$ hydrogenation**. Fig. 2 shows the performance of a set of cobalt catalysts in $CO_2$ hydrogenation with a feed gas at a pressure of 2.0 MPa containing $CO_2$ and $H_2$ ($H_2/CO_2 = 3:1$, molar). The products, besides methanol, were CO and methane, formed respectively by the reverse water-gas shift and methanation reactions. A cobalt catalyst without silica ($CoO_x$) was characterized under our conditions by a $CO_2$ conversion of 6.7%, with CO and methane as the dominant products, and a slight amount of methanol (Fig. 2a). Significantly, the inclusion of silica in the cobalt catalyst improved both the $CO_2$ conversion and methanol selectivity. For example, the $Co@Si_{0.52}$ catalyst gave $CO_2$ conversion and methanol selectivity of 9.0% and 47.9%, respectively. The methanol selectivity was further optimized by changing the cobalt/silica ratio, with the methanol selectivity of 70.5% at a $CO_2$ conversion of 8.6% for $Co@Si_{0.95}$ (Supplementary Table 1, Fig. 2b and c). In the catalytic reaction experiment, methanol was the sole carbon-containing liquid product (condensed in a cold trap downstream of the reactor) without any $C_{2+}$ by-products, which are usually formed in conversions with cobalt-containing catalysts[5,7], revealing a potentially valuable methanol production process.

In contrast, more silica in the catalyst led to decreased $CO_2$ conversions and lower methanol selectivity, illustrated by data characterizing the performance of $Co@Si_{1.48}$ and $Co@Si_{1.87}$, which might be due to changes in the state of cobalt and/or blocking of cobalt active sites by silica. In contrast, the conventional cobalt catalyst ($Co/SiO_2$) gave a $CO_2$ conversion of 7.3% and a methanol selectivity at 16.6%, with CO being the dominant product under the equivalent reaction conditions. These data confirm the unusual catalytic performance of $Co@Si_{0.95}$ in the $CO_2$ hydrogenation.

As expected, increased operating temperatures of the $Co@Si_{0.95}$ catalyst (Supplementary Fig. 4) gave higher conversions, with the methanol selectivity being >70% at 260-320 °C but decreasing at temperatures >320 °C. Similar trends were observed with the other $Co@Si_x$ catalysts (Supplementary Figs. 2-7). In these cases, the $Co@Si_x$ catalysts exhibited a marked decrease in selectivity to the undesired methane compared with the conventional cobalt catalysts (Supplementary Fig. 8). The conventional $Co/SiO_2$ was characterized by methanol selectivity generally <25% at temperatures in the range of 260-380 °C (Supplementary Figs. 9 and 10), where the $C_{2+}$ hydrocarbons were also detected with selectivity of 4.0%-8.5% at 260-380 °C. As shown in Fig. 2b, $Co@Si_{0.95}$ catalyst gave methanol productivity of 59.7 mmol $g_{cat}^{-1}$ $h^{-1}$, outperforming $Co/SiO_2$ and even the other supported copper and noble-metal catalysts that have been reported to be excellent for the $CO_2$-to-methanol transformation (Supplementary Table 3)[19,25,28]. For example, the methanol productivity of $Co@Si_{0.95}$ was found to be 10-fold greater than that of $Cu/SiO_2$ under comparable conditions[39].

The conventional supported metal nanoparticle catalysts generally suffer from the poor stability[29,30]. For example, the standard commercial $Cu/ZnO/Al_2O_3$ catalyst (Supplementary Figs. 11-13) for synthesis of methanol from $CO_2$ hydrogenation, evaluated in a wide temperature range (200-380 °C, Supplementary Fig. 12), gave the performances that are sensitive to the reaction temperatures. The best methanol yield appeared at

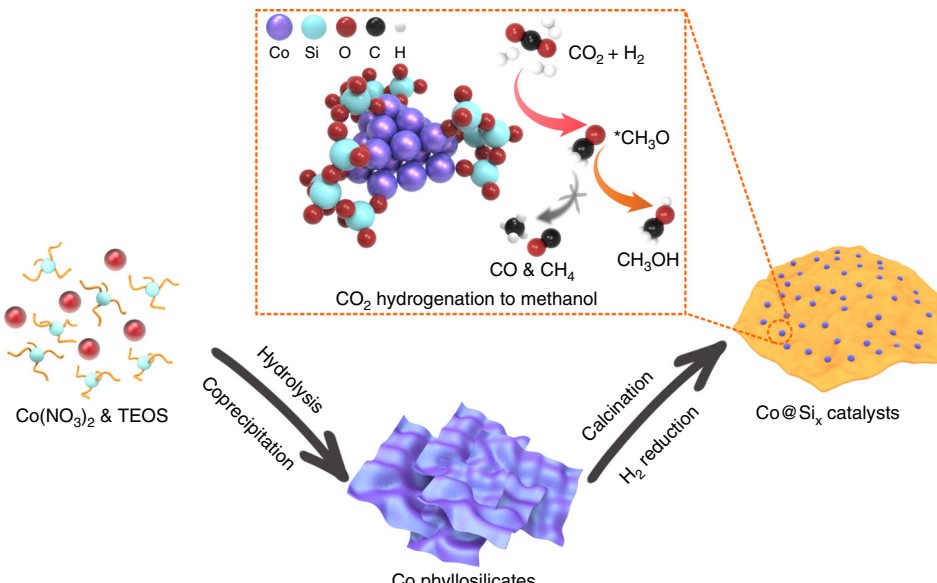

**Fig. 1 Synthesis and catalysis strategies of Co@Si$_x$ catalysts.** The procedures with cobalt phyllosilicates as intermediates for synthesizing Co@Si$_x$. Within the highlighted square, the CO$_2$-to-methanol transformation on Co@Si$_x$ catalysts.

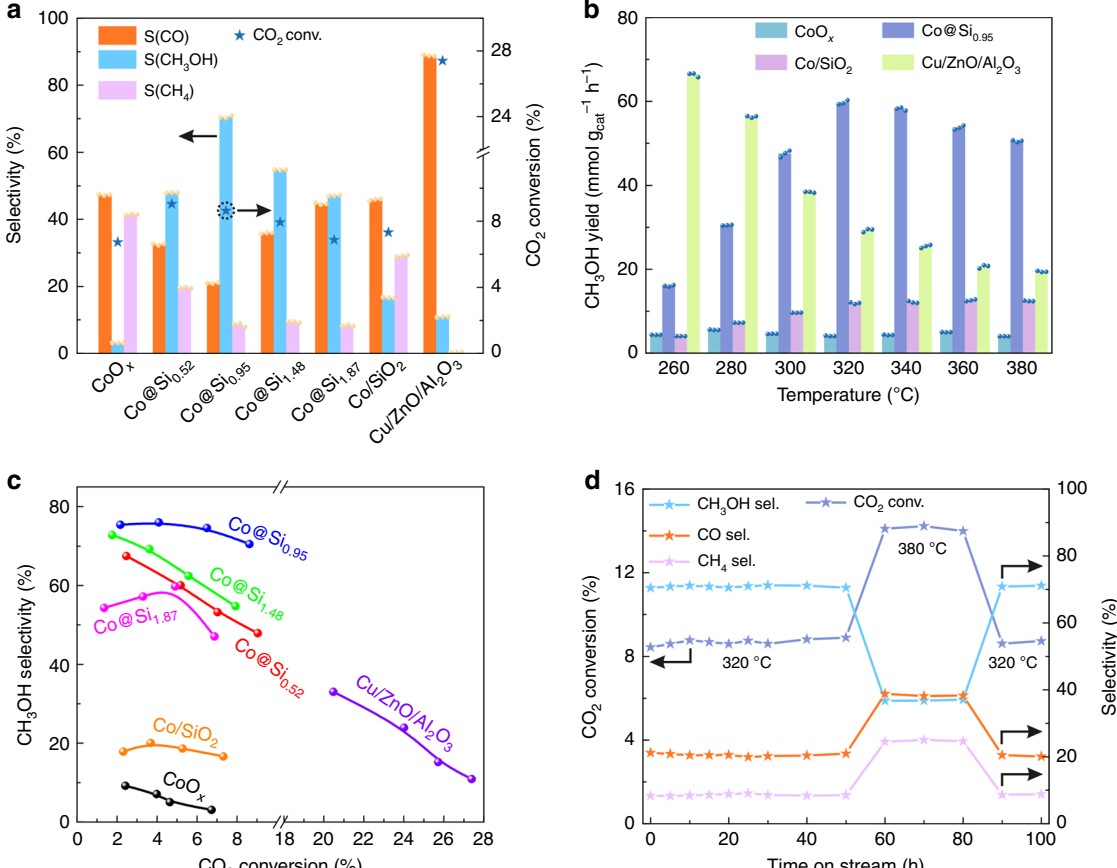

**Fig. 2 Performance of Co@Si$_x$ catalysts in CO$_2$ hydrogenation. a** Performance of catalysts, standard reaction conditions for Co@Si$_{0.95}$: 0.2 g of catalyst, pressure = 2.0 MPa; H$_2$/CO$_2$ feed ratio = 3:1, molar; temperature = 320 °C; GHSV = 6000 mL/g h. The mass of the other Co@Si$_x$ catalysts was chosen to give the same cobalt content in the reactor. **b** Dependences of methanol yield on temperature. Each reaction was performed three times, and the corresponding data points are provided in the bar charts. Error bounds for the conversion and selectivity are ± 0.3% and ± 0.5%, respectively. **c** CO$_2$ conversion and CH$_3$OH selectivity of Co@Si$_x$ catalysts at 260–320 °C. **d** Stability test of Co@Si$_{0.95}$ catalyst operated for 100 h in a flow reactor.

240 °C, giving productivity of 70.8 mmol $g_{cat}^{-1}$ $h^{-1}$ with $CO_2$ conversion of 15.2% and methanol selectivity of 47.6%, which is higher that of the Co@Si$_{0.95}$ catalyst (59.7 mmol $g_{cat}^{-1}$ $h^{-1}$). However, the Cu/ZnO/Al$_2$O$_3$ was characterized by a markedly inferior performance in the reaction life test, losing almost half of the methanol yield after reaction at 240 °C for 50 h (Supplementary Fig. 13). This result is in agreement with the knowledge of the Cu/ZnO/Al$_2$O$_3$ catalyst, whereby the Cu nanoparticles easily sinter into larger ones and cause deactivation[29,30]. Significantly, Co@Si$_{0.95}$ underwent almost negligible decay in the CO$_2$ conversion and methanol selectivity in 100 h of onstream operation (70 h at 320 °C and 30 h at 380 °C, Fig. 2d).

To the best of our knowledge, this excellent performance of Co@Si$_{0.95}$ catalyst in the CO$_2$ hydrogenation to methanol is unmatched. We are led to hypothesize that the silica support plays a key role, because the comparable silica-supported catalyst, Co/SiO$_2$, did not show this behavior. We were thus motivated to investigate the catalysts in depth and to determine catalytic structure–performance relationships.

**Catalyst structure study.** A transmission electron microscopy (TEM) image of Co@Si$_{0.95}$ (Fig. 3a) shows a lamellar structure of cobalt phyllosilicates. A high-angle annular dark field scanning transmission electron microscopy image (HAADF-STEM, Fig. 3b) and EDX elemental maps (Fig. 3c–e) demonstrate uniform dispersions of cobalt and silicon. The TEM image of Fig. 3f shows cobalt nanoparticles with an average diameter of 3.9 nm supported on the silica. A high-resolution TEM (HRTEM) image reveals the co-existence of metallic Co and CoO phases on the cobalt nanoparticles present in Co@Si$_{0.95}$ (Fig. 3g), which is further confirmed by the fast Fourier-transform (FFT) analysis (Fig. 3h) and XRD patterns. The cobalt nanoparticles on a series of Co@Si$_x$ samples have similar diameters, as evidenced by the HRTEM characterization. In contrast, the CoO$_x$ and Co/SiO$_2$ catalysts incorporate metallic Co as the dominant phase

(Supplementary Figs. 14–18). These data indicate a role of silica controlling the dispersion and the oxidation state of cobalt.

The cobalt–silica interaction on Co@Si$_x$ samples was investigated with FT-IR spectroscopy, with the bands at 665 and 1025 cm$^{-1}$, assigned to the Co-O-SiO$_n$ linkage (Fig. 3i and Supplementary Fig. 19)[40]. In contrast, these bands are undetectable in the FT-IR spectrum of Co/SiO$_2$, consistent with the lack of substantial interactions between cobalt and silica. X-ray absorption near edge structure (XANES) and extended X-ray absorption fine structure (EXAFS) spectra were recorded to characterize the oxidation states and coordination environments of Co in the Co@Si$_x$ samples. The Co K-edge XANES spectra of Co@Si$_x$ samples exhibit pre-edge features of the Co $1s$-$3d$ absorption transition at 7709.5 eV, with absorption edge positions of 7721.6 ± 0.2 eV (Fig. 3j and Supplementary Fig. 20)—these features are characteristic of cobalt oxides[41]. In contrast, the Co K-edge XANES of Co/SiO$_2$ is represented by an edge position of 7709.0 eV, assigned to metallic cobalt. These results point to the presence of cationic cobalt bonded to the silica, with Co-O-SiO$_n$ linkages at the Co-SiO$_2$ interfaces stabilizing the dispersed cobalt species in Co@Si$_x$. X-ray absorption spectra (XAS) recorded at the O K-edges of Co@Si$_x$ provide evidence confirming the hypothesis: the spectra include peaks assigned to Co-O bonds, at 532.5 and 539.8 eV[42], whereas the Co/SiO$_2$ exhibits an extremely weak Co-O signal because of its metallic feature (Supplementary Fig. 21).

In contrast, EXAFS spectra of Co/SiO$_2$, recorded at the Co K-edge (Fig. 3k), include a Co-Co shell with a distance determined in the fitting to be 2.50 5Å, with a coordination number of 9.3, indicating the dominant presence of metallic cobalt (Supplementary Table 4). Consistent with our interpretation, the Co-Co contributions characteristic of metallic cobalt are extremely weak in the spectra of Co@Si$_x$ samples. The EXAFS spectra indicate Co-O and Co-Co shells at distances of 2.05 and 3.02 Å, respectively, for Co@Si$_{0.95}$, with coordination numbers of 4.2 and 10.7, consistent with the presence of nonmetallic cobalt bonded to silica.

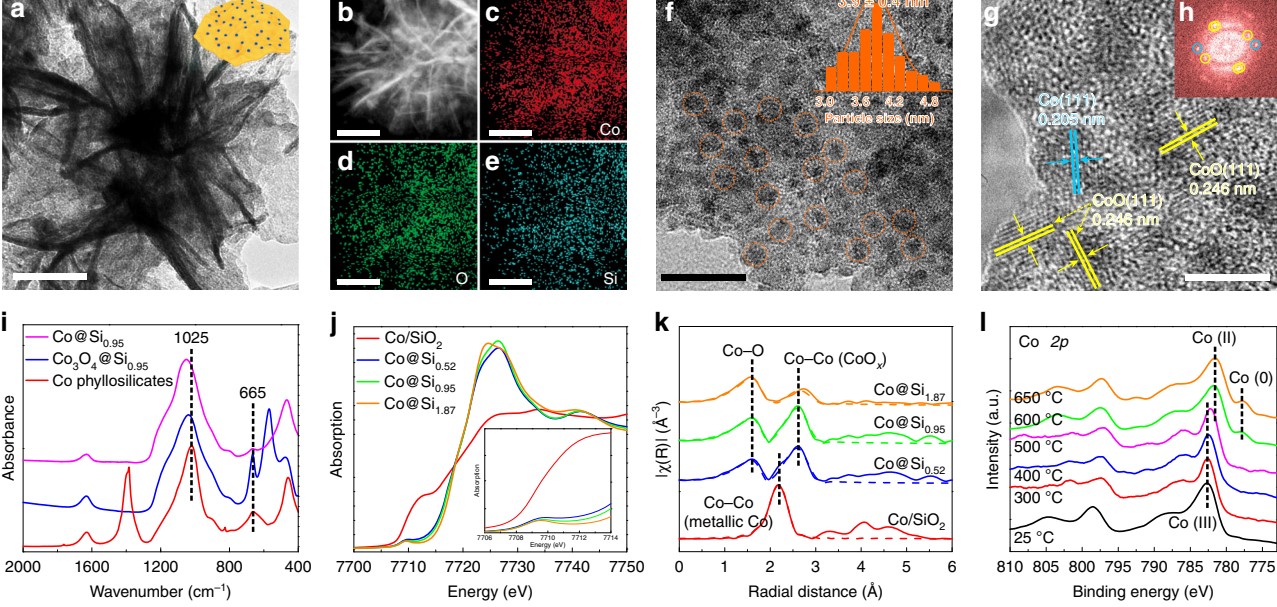

**Fig. 3 Structural characterization of Co@Si$_{0.95}$ catalyst. a** TEM image of Co@Si$_{0.95}$. **b–e, b** HAADF-STEM image and EDX elemental maps of **c** Co, **d** O, and **e** Si of Co@Si$_{0.95}$. **f, g** HRTEM images of Co@Si$_{0.95}$. The orange circles highlight the Co nanoparticles (inset of **f**: size distribution of Co nanoparticles.) **h** FFT image of the Co nanoparticles corresponding to the HRTEM image in **g**. Scale bars: 100 nm in **a–e**, 20 nm in **f**, and 5 nm in **g**. **i** FT-IR spectra of Co@Si$_{0.95}$ samples. **j, k** Co K-edge X-ray absorption spectra. **j** XANES spectra (inset: enlarged pre-edge region) and **k** EXAFS spectra with $k^2$-weighted data (solid line) and fit corresponding to recommended model (dashed line) of Co@Si$_x$ and Co/SiO$_2$ samples. **l** In situ Co $2p$ XPS spectra of Co$_3$O$_4$@Si$_{0.95}$ under 0.1 mbar of H$_2$ at various temperatures.

To further characterize these dispersed cobalt species, we did in situ Co $2p$ XPS experiments with the samples undergoing reductive treatments (Fig. 3l and Supplementary Figs. 22–25). The as-synthesized $Co_3O_4@Si_{0.95}$ sample was characterized by a dominant signal assigned to $Co^{3+}$ (782.4 eV, Supplementary Fig. 26), which was resistant to reduction and unchanged even after exposure to $H_2$ at 500 °C. Reduction at 600 and 650 °C gave spectra indicating the predominant presence of $Co^{2+}$ (781.5 eV) with some $Co^0$ (777.8 eV), indicating that the surface of $Co@Si_{0.95}$ incorporated predominantly cobalt oxide species and a small amount of metallic cobalt after vigorous reduction. Although the in situ XPS was performed using hydrogen with a lower pressure than that of the practical reduction treatment because of the XPS technique limitation[43], it is sufficient to reduce the cobalt species. For example, the $Co^{3+}$ on $Co_3O_4/SiO_2$ was easily reduced to $Co^{2+}$ at a temperature of only 300–400 °C, and $Co^0$ was the only cobalt species detected after reduction at 500 and 600 °C (Supplementary Figs. 27 and 28). This result is in good agreement with the $H_2$-TPR measurement of the cobalt oxide sample (Supplementary Fig. 29). These results all support the conclusion that the cobalt species in $Co@Si_x$ are strongly resistant to reduction.

In order to provide more evidence, we treated the $Co@Si_{0.95}$ sample with relatively high-pressure $H_2$ at 600 °C for 2 h (10% $H_2$ in Ar, 2 MPa), which should provide enough hydrogen for reducing the cobalt species. Significantly, the treated $Co@Si_{0.95}$ still contained cobalt oxide as the dominant phase with a small amount of metallic cobalt, as confirmed by the XRD (36.4°, 42.5° and 61.5° assigned to CoO phase) and XPS characterizations (781.5 eV assigned to $Co^{2+}$ and 777.8 eV assigned to $Co^0$) (Supplementary Fig. 30). The conclusion is further confirmed by in situ Raman spectra (Supplementary Fig. 31). By increasing reduction temperature to 600 °C, $Co@Si_x$ samples still showed a typical Raman signal of Co–O species, which was undetectable on the reduced $Co/SiO_2$ . Even after reaction for 100 h under the practical $CO_2$ hydrogenation conditions (Fig. 2d), the $Co@Si_{0.95}$ sample still exhibited the dominant CoO phase with a relatively small amount of metallic Co (XRD and XPS in Supplementary Fig. 32), confirming the difficult-to-reduce cobalt species on the $Co@Si_{0.95}$ catalyst, in good agreement with the in situ XPS investigation.

On the basis of these results, we propose that the silica influences the cobalt oxidation state, resulting in structures that are active and selective catalysts for methanol formation and not for methane and CO formation[35]. The relationships between the methanol yield in $CO_2$ hydrogenation and $Co^0/Co^{2+}$ ratio for various catalysts are presented in Supplementary Fig. 24f. Compared with $CoO_x$ catalyst, the $Co@Si_{0.52}$ and $Co@Si_{0.95}$ with $Co^{2+}$ species exhibited enhanced methanol yields. Further decreasing the $Co^0/Co^{2+}$ ratio reduced the methanol yields over $Co@Si_{1.48}$ and $Co@Si_{1.87}$ catalysts. These data confirm the balanced metallic Co and CoO phases on the catalysts are important for the methanol production. More $Co^0$ species cause the formation of a large amount of methane with poor methanol selectivity. Consistent with this picture, the hydrogen dissociation ability was evaluated by the catalysis in HD production by the reaction of $H_2$ with $D_2$ (a measurement of activity for activation of dihydrogen) over the $Co/SiO_2$ and $Co@Si_{0.95}$ catalysts. The product of the former contained 85% HD and only 27% HD for the latter (Supplementary Fig. 33), suggesting the $H_2$ dissociation ability of $Co@Si_{0.95}$ was weakened because such ability was strongly related to the metallic Co. The surprising finding is that the $Co@Si_{0.95}$ with lower $H_2$ dissociation ability even exhibits higher $CO_2$ conversions than the $Co/SiO_2$ catalysts with high activity in $H_2$ activation. The sole CoO phase is known to have poor activity for the hydrogenation. Therefore, the $Co@Si_{0.95}$ catalyst with balanced phases exhibited the best performance

among these samples (Supplementary Figs. 34–36). Apart from influencing the cobalt oxidation state, more silica species might block more surface sites of the $Co@Si_x$ catalysts, which would also influence the catalytic performance. These data might explain why the various $Co@Si_x$ catalysts with similar cobalt nanoparticle sizes have markedly different catalytic performance.

We conclud that the silica acts as an effective support for turning the cobalt nanoparticles from catalysts for methanation/ CO formation into catalysts for methanol production, exhibiting simultaneously high activity, selectivity, and durability for the $CO_2$-to-methanol transformation. Such different catalytic features compared with the conventionally supported cobalt catalysts are associated with the Co-O-$SiO_n$ linkage. It is reasonable to understand this linkage stabilizes the cobalt nanoparticles and hinders the sintering during the calcination/reduction/reaction under harsh conditions. For example, after reaction for 100 h, the used $Co@Si_{0.95}$ catalyst still incorporated the cobalt nanoparticles with an average diameter of 3.9 nm, which is almost unchanged compared with the as-synthesized catalyst (Supplementary Fig. 37). The $Co_2C/CoC$ species are undetectable on $Co@Si_{0.95}$ as confirmed by the XRD pattern and HRTEM images (Supplementary Figs. 32 and 37). In contrast, the used $Co/SiO_2$ contained predominantly metallic Co accompanied by $Co_2C$ species (Supplementary Figs. 38 and 39) after the equivalent test for 100 h, in good agreement with expectation[44,45]. The remarkably different phenomena of $Co@Si_{0.95}$ compared with the conventional cobalt catalysts are attributed to the Co-O-Si linkage on the $Co@Si_{0.95}$ catalyst, which hindered the carbonization of cobalt species[46,47].

**Mechanism study.** In order to gain insight into how the silica modification influences the reaction pathways, we characterized the samples using IR spectroscopy in $CO_2$ adsorption and hydrogenation. Supplementary Fig. 40 shows the spectra of various catalysts after exposure to $CO_2$, with the $CoO_x$ characterized by bands at 1260, 1530, 2850, 2945, and 3015 cm$^{-1}$, assigned to carboxylate ($CO_2^{\delta-}$, 1260 cm$^{-1}$), formate (*HCOO, 1530, 2850, and 2945 cm$^{-1}$), and *$CH_x$ species (3015 cm$^{-1}$), respectively[28,48–50]. The $CO_2^{\delta-}$ is from the chemisorbed $CO_2$ species on the cobalt sites, and the *HCOO and *$CH_x$ are from the interaction of chemisorbed $CO_2$ with hydrogen adatom on cobalt sites resulted from the $H_2$ pretreatment. The *$CH_x$ species, which are known intermediates in methane formation, confirm that deep hydrogenation occurs on the $CoO_x$ catalyst[50]. It is significant that the *$CH_x$ band (3015 cm$^{-1}$) was almost undetectable in the spectra of the $Co@Si_x$ catalysts, consistent with the suppression of deep hydrogenation of $CO_2$ which requires metallic sites[35]. The spectra further show that more silica species in $Co@Si_x$ correspond to lower intensity of *HCOO (2850 and 2945 cm$^{-1}$), also being correlated with those of the chemisorbed $CO_2$ ($CO_2^{\delta-}$, 1244–1276 cm$^{-1}$).

To identify reaction intermediates, we collected in situ DRIFTS spectra (Fig. 4a and Supplementary Figs. 41–43), bringing the catalysts in contact with feed gases having varied $CO_2$ and $H_2$ concentration at 350 °C. Exposure of $Co@Si_{0.95}$ to $CO_2$ without $H_2$ gave rise to bands, mainly including those of $CO_2^{\delta-}$ (1246, 1592 cm$^{-1}$), $CO_3^{2-}$ (1435 cm$^{-1}$), and *HCOO (1337, 2850, 2945 cm$^{-1}$)[28]. When $H_2$ was present ($CO_2$:$H_2$, molar ratio = 3), the bands of $CO_2^{\delta-}$ (1246, 1592 cm$^{-1}$) were markedly weakened and those of *HCOO (1360, 1560 cm$^{-1}$) enhanced. Simultaneously, new bands appeared at 1048, 1462, 2830, and 2928 cm$^{-1}$, assigned to *$CH_3O$ species. Continuous feeding of $H_2$ (switch off $CO_2$) markedly increased the *HCOO and *$CH_3O$ band intensities (Fig. 4b, 0–12 min). After 12 min, the *HCOO signal was constant, but the *$CH_3O$ signal continued

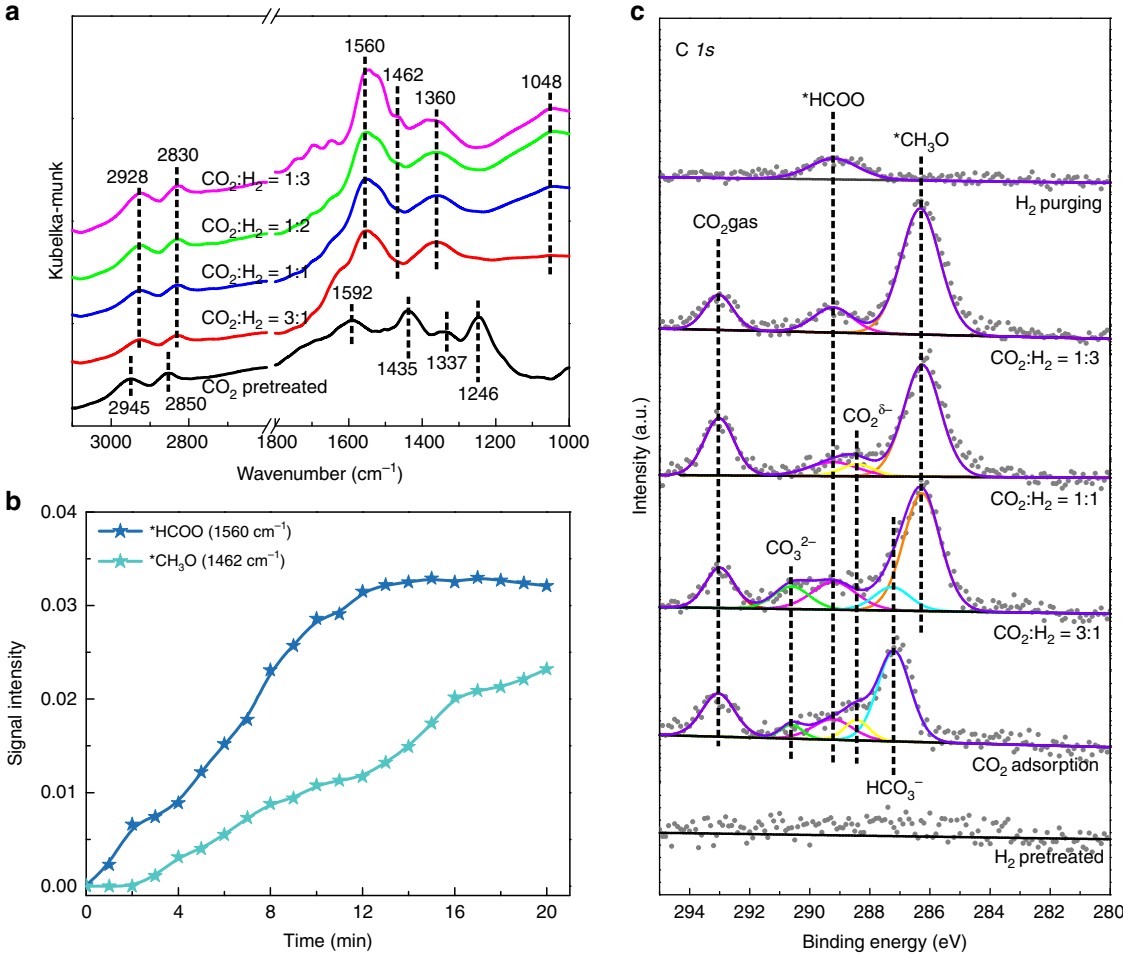

**Fig. 4 Mechanism of CO₂ hydrogenation on Co@Si₀.₉₅ catalyst. a** In situ DRIFTS spectra of Co@Si₀.₉₅ catalyst at 350 °C in contact with CO₂ + H₂. **b** Time-dependent DRIFTS band intensities characterizing surface *HCOO and *CH₃O species during the CO₂ + H₂ reaction on Co@Si₀.₉₅ at 350 °C. **c** In situ C *1s* XPS spectra of Co@Si₀.₉₅ in contact with 1.2 mbar of CO₂ + H₂ atmosphere with controlled ratios at 250 °C.

to increase. In contrast with the spectra of $Co@Si_{0.95}$, the comparable spectra $Co/SiO_2$ and $CoO_x$ give evidence of only trace of $*CH_3O$ (in contrast to the stronger bands of $*HCOO$ and/or $*CH_x$ species). The $*CH_3O$ species are readily converted into methanol by hydrogenation[22,24], and the high methanol selectivity of $Co@Si_{0.95}$ is ascribed to these species as reaction intermediates. The trace of $*CH_3O$ signal on $Co/SiO_2$ and $CoO_x$ is consistent with their low methanol selectivity.

Further investigation of the reaction intermediates on $Co@Si_{0.95}$ was performed with ambient-pressure (AP) XPS. Although the $CO_2$ and $H_2$ pressure was much lower than that in the practical tests, it is sufficient to react with the catalyst surface. Changes in the surface and reaction intermediates were shown by X-ray photoelectron spectra[16,27,50]. $CO_2$ is readily adsorbed on this catalyst, giving rise to C *1s* bands at 293.0, 290.6, 289.2, 288.4, and 287.2 eV, assigned to gaseous $CO_2$, $CO_3^{2-}$, $*HCOO$, $CO_2^{\delta-}$, and $HCO_3^-$ species, respectively (Fig. 4c)[51]. When the sample was exposed to $H_2$ ($CO_2$:$H_2$, molar ratio = 3), signals characteristic of $CO_2^{\delta-}$ and $HCO_3^-$ were reduced and that of $*HCOO$ enhanced. Concomitantly, a signal appeared at 286.3 eV and became dominant, indicating the formation of abundant $*CH_3O$ species. More $H_2$ in the feed gas ($CO_2/H_2$ ratio = 1/3, molar) markedly reduced the bands of chemisorbed $CO_2$ ($CO_2^{\delta-}$ and $CO_3^{2-}$), which were quickly transformed to $*CH_3O$ species by feeding sufficient hydrogen, whereas the signal of $*HCOO$ remained almost unchanged. When the feed gas was switched

to pure $H_2$ without $CO_2$, the $*CH_3O$ signal disappeared immediately—this species was evidently further hydrogenated to form methanol. However, the $*HCOO$ signal remained essentially constant, as this species was resistant to hydrogenation on the catalyst (Supplementary Figs. 44 and 45). In contrast, the $Co/SiO_2$ catalyst was also characterized by chemisorbed $CO_2$, but with extremely weak $*CH_3O$ bands under the equivalent conditions (Supplementary Fig. 46), in good agreement with the DRIFTS spectra. These data confirm the importance of the silica-supported species containing cationic cobalt for $*CH_3O$ formation and stabilization, even when the reaction atmosphere contains only little $H_2$ ($CO_2/H_2$, molar ratio = 10:1, Supplementary Figs. 47 and 48).

The easily detected abundant $*CH_3O$ signals in the in situ DRIFTS and XPS characterization confirm the fast formation and slow further transformation of $*CH_3O$ on the $Co@Si_x$ catalyst (Supplementary Figs. 49 and 50). Apart from the hydrogenation to $*CH_3OH$, the $*CH_3O$ species might also undergo C–O cleavage and the subsequent hydrogenation to $CH_4$[24,28,52], as well as the dehydrogenation to CO. With regard to the $*CH_3O$ transformation, multiple reaction pathways have been proposed in the formation of $*CH_x$ intermediates, that are ready to proceed the methanation[50]. In this route, the C–O cleavage is always regarded to be the rate controlling step[24,28,52]. Reported density functional theoretical calculations have revealed that the cleavage of the C–O bond in $*CH_3O$ requires the metallic Co surface or the CoO

surface with abundant oxygen vacancies. The CoO(100) surface saturated with oxygen leads to a high energy barrier for the *CH$_3$O dissociation at 2.71 eV [1.45 eV for Co(111) surface and 1.01 eV for the oxygen vacancy-rich CoO(100)][35]. The Co@Si$_{0.95}$ catalyst with not-easy-to-reduce oxygen species provided an ideal catalyst surface for hindering the C–O cleavage. In addition, the C–O bond cleavage is known to be assisted by hydrogen[53], and the relatively lower activity of Co@Si$_{0.95}$ for dihydrogen activation (Supplementary Fig. 33) might also contribute to stabilization of *CH$_3$O intermediates to avoid C–O cleavage. Apart from the *CH$_3$O decomposition, another possible route for methane or other higher hydrocarbons formation is via the direct CO dissociation into *C species, which has been experimentally and theoretically studied in the cobalt-catalyzed Fischer–Tropsch synthesis[31,54]. Metallic cobalt and cobalt carbide were found to be efficient for CO dissociation, but the oxidized cobalt surface is known to be less active, which is also confirmed by the poor activity of Co@Si$_{0.95}$ in the CO hydrogenation (CO conversion of 0.7% and methanol selectivity of 22.7%) under the employed reaction conditions (360 °C, 2.0 MPa, Supplementary Fig. 51).

In addition to the C–O cleavage, another possible route for *CH$_3$O transformation is dehydrogenation, giving CO product. To probe this reaction, we performed temperature-programmed surface reaction (TPSR) experiments to evaluate the reaction of *CH$_3$O on different catalysts, with methanol as a feed because it easily forms *CH$_3$O species. As shown in Supplementary Fig. 52, the CO signal centered at 280 °C characterizing the Co/SiO$_2$ catalyst demonstrates the dehydrogenation of the *CH$_3$O species indeed occurred on the surface of metallic cobalt. In contrast, no CO signal was observed in equivalent experiments with the Co@Si$_{0.95}$ catalyst at temperatures <300 °C, evidencing the enhanced ability of Co@Si$_{0.95}$ catalyst to resist dehydrogenation. In addition to the CO, the methane signal was detected at 340 °C on Co/SiO$_2$ catalyst, attributed to the C–O dissociation and deep hydrogenation to methane. In contrast, no methane signal was observed on Co@Si$_{0.95}$ catalyst, even at temperature up to 420 °C. These results might explain the reduced methanation and CO formation on the Co@Si$_{0.95}$ catalyst, whereby the stabilization of *CH$_3$O species on the catalyst surface hinders the C–O cleavage and deep dehydrogenation. This feature contributes to the high methanol productivity via further hydrogenation of the *CH$_3$O intermediates, in good agreement with the XPS results (Supplementary Figs. 47c and 48c).

## Discussion

A central result emerging from the in situ DRIFTS and XPS data is that the *CH$_3$O species on Co@Si$_{0.95}$ act as intermediates for methanol formation. The observation of abundant *CH$_3$O species indicates that they are stable intermediates. The results suggest that the CO$_2$ hydrogenation on Co@Si$_{0.95}$ might proceed by a mechanism similar to that occurring on the well-known Cu/ZnO catalyst[24], whereby the transformation of *CH$_3$O is crucial for the selective formation of methanol. Another central result is the catalyst performance data showing that methanol forms with much less accompanying CO and methane—their formation from *CH$_3$O would require deep dehydrogenation and breaking of the C–O bond, respectively, which readily occurs on metallic cobalt but not on the cobalt oxide surface with unreducible oxygen, according to the reported simulation results[35]. Thus, we infer that the dominant cobalt oxide phase on Co@Si$_{0.95}$ provides a nearly optimum structure for hindering the side reactions and facilitating methanol formation.

Catalysts in this class offer a compelling example showing the key role of a nominally inert support—silica—turning cobalt from a nonselective catalyst into highly selective catalyst for methanol

production. We suggest this work may open the way to new control of catalysts by supports and help guide the design of improved catalysts for selective hydrogenation of CO$_2$.

## Methods

**Materials**. Co(NO$_3$)$_2$·6H$_2$O (99.0%), Co$_3$O$_4$ (99.5%, 100 nm), CO(NH$_2$)$_2$ (99.5%), and tetraethylorthosilicate (TEOS, 99.0%) were obtained from Aladdin Chemical Reagent Company. NaOH (96.0%), NH$_3$·H$_2$O (25.0%–28.0%) and amorphous SiO$_2$ were obtained from Shanghai Lingfeng Chemical Reagent Co. Ltd. Cu/ZnO/Al$_2$O$_3$ was provided by Beijing Sanju Environmental Protection & New Materials Co. Ltd. Pure Ar, CO, CO$_2$, CH$_4$, 10% H$_2$/Ar, 10% CO$_2$/Ar, CO$_2$/H$_2$/Ar (25%/50%/25%, 20%/60%/20%, and 19%/76%/5%) and CO/H$_2$/Ar (30%/60%/10%) were provided by Hangzhou Jingong Special Gases Co. Ltd.

**Catalysts preparation**. Synthesis of Co$_3$O$_4$@Si$_{0.95}$ and Co@Si$_{0.95}$ catalysts: The Co(NO$_3$)$_2$·6H$_2$O (40 mmol) and TEOS (40 mmol) were dissolved in 200 mL mixed liquor containing water and ethanol with the volume ratio of 3/1, followed by adding 20 mL of NH$_3$·H$_2$O. After stirring at room temperature for another 8 h, the precipitate was separated by filtration, washed with deionized water, and dried at 100 °C overnight to obtain Co phyllosilicates. The Co$_3$O$_4$@Si$_{0.95}$ was obtained by calcining the Co phyllosilicates at 500 °C in air for 4 h. After reducing Co$_3$O$_4$@Si$_{0.95}$ in flowing hydrogen (10% H$_2$/Ar, 60 mL/min) for 3 h at 600 °C, the Co@Si$_{0.95}$ catalyst was obtained.

Synthesis of Co$_3$O$_4$@Si$_x$ and Co@Si$_x$ catalysts with different Si/Co ratios: The Si/Co ratio was adjusted to obtain a series of Co@Si$_x$ catalysts, where $x$ is the Si/Co ratio. The Co@Si$_x$ catalysts with different initial Si/Co ratios of 0.52, 1.48, and 1.87 were synthesized by procedures similar to those used for Co$_3$O$_4$@Si$_{0.95}$ and Co@Si$_{0.95}$ catalysts except for changing the amount of TEOS to 20, 60, and 80 mmol.

Synthesis of CoO$_x$ catalyst: The CoO$_x$ catalyst was synthesized following the similar synthesis procedures for Co@Si$_x$ catalysts without using TEOS.

Synthesis of Co/SiO$_2$ catalyst: 1.2 g of SiO$_2$ was dispersed into 100 mL of aqueous solution containing 20 mmol of Co(NO$_3$)$_2$·6H$_2$O and 100 mmol of CO(NH$_2$)$_2$, followed by stirring at 80 °C for 4 h, then the precipitate was separated by filtration and washed with deionized water. After drying at 100 °C overnight, calcining at 400 °C in air for 4 h and reducing in flowing hydrogen (10% H$_2$/Ar, 60 mL/min) for 3 h at 600 °C, the Co/SiO$_2$ catalyst was obtained.

Synthesis of Co@Si$_{0.95}$-Na catalyst: 1.0 g of Co@Si$_{0.95}$ catalyst was dispersed into 100 mL of aqueous solution containing 1 mmol of NaOH, followed by stirring at room temperature for 3 h. Then the catalyst was separated by filtration and washed with deionized water. After drying at 100 °C overnight and reducing in flowing hydrogen (10% H$_2$/Ar, 60 mL/min) for 3 h at 600 °C, the Co@Si$_{0.95}$-Na catalyst was obtained.

**Characterization**. X-ray diffraction (XRD) patterns were collected on a Rigaku D/MAX 2550 diffract meter with Cu Kα radiation (λ = 1.5406 Å). The Fourier-transform IR (FT-IR) spectra were recorded on a Bruker Vector 22 in the range of 4000–400 cm$^{-1}$. The composition of Co@Si$_x$ catalysts was measured by an inductively coupled plasma (ICP) analysis (Perkin-Elmer 3300DV). Transmission electron microscopy (TEM), scanning transmission electron microscopy (STEM) images and selected area electron diffraction (SAED) were obtained on a JEM-2100F electron microscopy with an acceleration voltage of 200 kV. X-ray photo-electron spectra (XPS) of the samples were recorded using a Kratos AXIS SUPRA with Al Kα X-ray radiation as the X-ray source. The binding energies were calibrated on the basis of the C 1s (284.8 eV) peak. X-Ray absorption near edge structure (XANES) and extended X-ray absorption fine structure (EXAFS) measurements at the Co K-edge were made at beamline 8-ID at the National Synchrotron Light Source II (NSLS II) at Brookhaven National Laboratory and BL14W1 beamline of the Shanghai Synchrotron Radiation Facility. The O K-edge soft X-ray absorption spectra (XAS) was measured at the BL12B-a beamline of the National Synchrotron Radiation Laboratory (NSRL). H$_2$-temperature-programmed reduction (H$_2$-TPR) was performed with a Finesorb-3010.

**In situ DRIFTS characterization**. DRIFTS were recorded using a Thermo Fisher Nicolet iS50 FT-IR spectrometer equipped with a MCT/A detector and ZnSe windows and a high temperature reaction chamber under ambient pressure. In a typical run, 50 mg of solid sample was loaded into the chamber and pretreated at 200 °C for 30 min in flowing Ar (20 mL/min). Then, the chamber was adjusted to the desired temperature (250 °C), and CO$_2$ (10% CO$_2$ in Ar) was flowed through the sample for 30 min. After removing the physically adsorbed CO$_2$ by pure Ar gas, the DRIFTS signals were recorded (Supplementary Fig. 40).

In order to observe the reaction intermediates on the catalyst surface, the similar procedures were repeated except using mixed gas of CO$_2$ and H$_2$. CO$_2$ (10% CO$_2$ in Ar) and H$_2$ (10% H$_2$ in Ar) with controlled ratios were continuously introduced to the chamber (40 mL/min) at 350 °C, and the data were collected (Figs. 4a, b and Supplementary Figs. 41–43).

**In situ Raman characterization**. Raman spectra were recorded using a HR800 Raman spectrometer equipped with an Ar excitation source ($\lambda = 514.532$ nm). The hydrogen was introduced into the sample chamber (10% $H_2$ in Ar) to reduce the solid samples at desired temperatures (25-600 °C) for 30 min, then the spectra were collected (Supplementary Fig. 31). For investigating the $CO_2$ adsorption, the samples were pretreated with $H_2$ at 250 °C and then the feed gases were introduced. For investigating the reaction on the sample, the above-mentioned procedures were repeated except using mixed gas of $CO_2$ and $H_2$ (1:3) in the treatment at 250 °C (Supplementary Fig. 45).

**In situ XPS characterization**. XPS spectra were recorded using a SPECS NAP-XPS with a monochromatic Al Kα source. The exposure to reaction gas was done by backfilling the NAP-XPS chamber. The binding energies were calibrated on the basis of the C 1s (284.8 eV) peak. In a typical run, 50 mg of solid sample was molded in advance and fixed in the chamber, then the sample chamber was evacuated. The blank XPS spectra were collected at 25 °C, followed by reducing the solid samples at controlled temperatures (300, 400, 500, 600, and 650 °C) in a hydrogen atmosphere (pure $H_2$, 0.1 mbar) for 10 min, then the data were collected to identify the changes of Co and O (Fig. 3l and Supplementary Figs. 23, 27 and 28).

For investigating the $CO_2$ adsorption on the samples, the chamber was vacuumed again to eliminate the excess hydrogen, and another feed gas (pure $CO_2$, 1.0 mbar) was introduced for 10 min at 250 °C, the XPS spectra were recorded in the meanwhile. For investigating the $CO_2$ hydrogenation reaction on the samples, the above-mentioned procedures were repeated except using mixed gas of $CO_2$ and $H_2$ with the desired gas ratio ($CO_2$:$H_2$ = 3:1, 1:1 and 1:3, total pressure was 1.2 mbar) in the treatment at 250 °C (Fig. 4c and Supplementary Figs. 44 and 46). In the end, 1.0 mbar of hydrogen was introduced to regain a fresh sample. The XPS spectra were recorded following the above-mentioned procedures. The $CO_2$ hydrogenation activity of the catalysts was further studied at a low hydrogen pressure. A mixed gas containing 0.1 mbar of hydrogen and 1.0 mbar of $CO_2$ was introduced into the chamber for 5 min. Then the gas was switched off and slowly evacuated from the chamber, and the XPS spectra were recorded (Supplementary Figs. 47 and 48).

**$CO_2$ hydrogenation**. The $CO_2$ hydrogenation was carried out in a tubular fixed-bed continuous-flow reactor equipped with gas chromatography (GC). 0.2 g of catalyst (40-60 mesh) was diluted with 0.4 g of quartz sand (40-60 mesh) in the catalyst bed. The reaction was conducted under reaction conditions of 1.0–4.0 MPa, 260–380 °C, $V(H_2:CO_2:Ar)$ = 50:25:25, 60:20:20, or 76:19:5, GHSV = 3000–12,000 mL/g h. The emission gas (Ar, CO, $CH_4$, $CO_2$, and $C_{2+}$ hydrocarbons) from the reactor was maintained at 130 °C and immediately transported to the sample valve of a Fu Li-9790 GC equipped with a thermal conductivity detector (TCD) and a Fu Li-9790 GC equipped with a flame ionization detector (FID). The liquid phase products ($CH_3OH$) were collected in a cold trap and then analyzed with a Fu Li-9790 GC equipped with FID, with benzyl alcohol as an internal standard. Error bounds for the conversion and selectivity are ±0.3% and ±0.5%, respectively.

## Data availability

The source data underlying Figs. 2–4 are provided as a Source Data file. The other primary data that support the plots within this paper and findings of this study are available from the corresponding author on reasonable request.

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

## Acknowledgements

This work was supported by the National Key Research and Development Program of China (2018YFB0604801), National Natural Science Foundation of China (21822203, 91645105, and 91634201), Natural Science Foundation of Zhejiang Province (LR18B030002), and the Fundamental Research Funds for the Central Universities. The work at the University of California was supported by the U.S. Department of Energy (DOE), Office of Science, Basic Energy Sciences (BES) Grant FG02-04ER15513. We acknowledge Beamline BL14W1 (Shanghai Synchrotron Radiation Facility), NSRL Beamline BL12B-a (National Synchrotron Radiation Laboratory), and Nano-X (SINANO, CAS). We thank Fang Chen for kindly helping with TEM characterization.

## Author contributions

L.X.W. performed the catalyst preparation, characterization, and catalytic tests. E.G. and B.C.G. did the EXAFS/XANES experiments and analysis. Y.W. participated the catalyst preparation and characterization. Z.G. and Y.C. performed in the XPS investigation. X.M. provided helpful discussions. L.W. and F.-S.X. designed this study, analyzed the data, and wrote the paper.

## Competing interests

The authors declare no competing interests.
