## [Peer Review File · Nature Communications]

Reviewers' comments:

Reviewer #1 (Remarks to the Author):

This article proposed that the selectivity of CO₂ hydrogenation was dependent on the oxidation states of Co. Metallic Co caused the hydrogenation of CO₂ to CO. The oxidized Co in Co-O-Si center caused the hydrogenation of CO₂ to methanol. This is rather interesting. However, several major concerns still exist. This work cannot be accepted unless these issues are well addressed.

a-There are obvious differences between in-situ XPS atmosphere treatment pressure conditions (pure H₂, 0.1 mbar; pure CO₂, 1.0 mbar; CO₂: H₂ = 3:1, 1:1 and 1:3, total pressure was 1.2 mbar, see SI in-situ XPS characterization for details) and the actual catalytic conditions (under reaction conditions of 1.0 to 4.0 MPa, see SiCO₂ hydrogenation for details). Can these in-situ XPS data reflect actual catalytic properties accurately under such different conditions? Under 1.0 mbar for in-situ XPS treatment, it is very likely that oxidized Co species are not completely reduced. However, under the actual catalytic conditions, the partial pressure of H₂ was much higher, so that Co might be completely reduced.

b-The published literatures show that Co⁰ tends to show high methane selectivity in CO₂ hydrogenation (Journal of CO₂ Utilization, 2018, 26: 221-229). In this article, Co/SiO₂ was synthesized as the comparison sample. A series of characterizations shows that Co in Co/SiO₂ was basically at metallic states. However, in this article, the major product for Co/SiO₂ was CO (see Figure 2a). Why did the selectivity of metallic Co in this work differ from those previously reported?

c-Metallic Co is a conventional FTS active site which can catalyze the hydrogenation of CO to long chain products. For Co/SiO₂, why were the produced CO molecules not hydrogenated into C₂+ products?

d-In terms of reaction mechanism, why can the stabilization of CH₃O* improve methanol selectivity? In my opinion, higher methanol selectivity and lower CO selectivity are mainly due to HCOO* intermediate species. According to the review (Nanoscale, 2015, 7, 8663), HCOO* pathway does not produce CO. In addition, if the adsorption of CH₃O* is strong, will methanol be overhydrogenated to CH₄? TPSR-MS experiment is suggested to find out from what temperature methanol will be hydrogenated to methane.

e-In this article, the authors only provided XRD data of different catalysts before and after reduction treatment. The comparison of catalysts structure and phase composition before and after reaction is not included. Then, will Co in Co@Six be partially carbonized after a long run?

f-In my opinion, when comparing the performance of different catalysts, the optimal conditions for each sample should be chosen. For Cu/ZnO/Al₂O₃, it is generally considered that the optimal operation temperature should be lower than 300 °C (Science, 2012, 336, 893-897). In this paper, the performance was compared at 320 °C, which was higher than the optimum temperature of Cu/ZnO/Al₂O₃. In addition, the performance of Cu/ZnO/Al₂O₃ used in this paper was too poor, as the conversion was only 1% at 200 °C, and methanol selectivity was just 60%. Compared with the data of Cu/ZnO/Al₂O₃ (conversion=~8%, methanol selectivity=~80%, at 200 °C) in Science Advances, 2017, 3: e1701290. The authors can directly use commercial Cu/ZnO/Al₂O₃ as comparison.

g-In the synthesis process of Co@Six, no surface treatment (such as adding TCAC as coating agent) was carried out on the particles obtained by coprecipitation to control the uniform coating of precipitated particles. Theoretically, the catalyst obtained by this method contained both Co coated by SiO_n and Co loaded on the surface. There should be differences in catalytic activity between these two Co sites. I suggest the authors to provide further evidences to clarify the distribution of Co centers in Co@Six and the contribution of different centers to the overall catalytic activity.

h-The electron micrographs and a series of spectral characterizations provided in this article did not reflect the difference between Co centers in Co@Six with different x values. However, when x changed from 0.52 to 1.58, the selectivity of Co@Six changed significantly (Figure 2a). Why was the selectivity dependent on the x value?

i-According to the description in the part of SI In-situ FT-IR characterization (IR spectra were recorded using a Thermo Fisher Nicolet iS50 FT-IR spectrometer equipped with a MCT / a detector), the author used the Drifts mode instead of FT-IR mode when collecting infrared data. The ordinate in the infrared spectrum of this article is marked incorrectly. For the Drifts spectrum, the ordinate is not "absorption". In addition, in this part of the description of the infrared test experiment, the pressure data during the test was not clearly provided. Were there any differences between the experiment condition and catalytic conditions?

j-For revealing the reaction path, I suggest the authors to conduct pressure-dependent tests to figure out the reaction orders.

Reviewer #2 (Remarks to the Author):

This work is concerned with the conversion of CO₂ to methanol over a silica modulated cobalt catalysts. The novelty of this work is the development of a stable, highly active methanol synthesis cobalt catalysts via the formation of Co phyllosilicates intermediates. This work used a judicious number of characterization techniques, both in situ and ex situ, to support the structure/activity claims. This work can potentially help to pave the way for the use of modulated support species on other transition metals to control the extent of hydrogenation.

Overall, I could recommended this work for publication with major revisions addressing the following concerns.

1. While the Co₃O₄@Six and Co@Six were rigorously characterized, there was no characterization of the base SiO_x material using the same preparation method. Furthermore, the surface areas of these catalysts were not explored. The surface area and overall porosity of the different Co@Six could have a considerable influence on the performance, where the influence of mass transfer was only briefly discussed for the high Co/Si samples.
2. The XANES spectra shown in Figure 2j should be normalized to an edge step of 1 such that comparisons between the sample can be readily accessed. An inset of the pre-edge region should be provided in the supplemental information that illustrates the differences in the pre edge regime to help guide the reader.
3. The EXAFS spectra in Figure 2k was fitted using single scattering paths from Co-O and Co-Co (from a CoO reference) and a Co-Co (from a metallic Co reference) which gave rise to reasonable radius for the specified paths. However, the authors should compare the direct XANES measurements with the XPS of the samples before in situ reductive pretreatment and determine if the result are in agreement. The comparison of the Co 2p spectra for all of the Co@Six samples should be added in a compiled supplemental figure so that the reader can understand the starting surface composition of the catalysts before exposure to hydrogen or reaction conditions.
4. The authors observed noticeably different reaction intermediates on the Co@Si_{0.95} catalyst relative to the control Co/SiO₂ sample, noticeably the formation of the *CH₃O intermediate which was verified via both in situ DRIFTS and XPS. The argument for the difference in reaction mechanism would be strengthened via the inclusion of apparent activation energies for the catalysts.

5. The authors attribute the increase in catalytic performance to the formation of Co-O-SiOn linkages that serve to stabilize the active CoO species, however, all of the Co@Six catalysts have constant particle size of ~3.8 nm. Given that the cobalt particle sizes are uniform within error, it would be reasonable to assume that the extent of interfacial Co-O-Si linkages should be constant throughout the catalyst. Therefore, the authors should give a more detailed description of the influence of the Co/Si ratio on the active site, where the increase in Co/Si ratio should result in higher mass transfer limitations.

Minor Comments:

1. The XRD patterns on Figure S1 should be normalized to show crystalline structure of the catalysts
2. The writing has several grammatical errors and should be carefully edited and proofread for clarity.
3. The supplemental information should have a table of content to guide the reader to the relevant information, given the length of the document.
4. Was any surface enrichment of the cobalt species for the Co@Six observed for these materials?
5. The EXAFS radius values reported in Table S3 should be truncated to three significant figures (i.e. 2.05 instead of 2.054) given the accuracy of most XAFS modeling software is at best 0.01 Å

Reviewer #3 (Remarks to the Author):

This paper reported a novel Co-based catalyst for CO₂ hydrogenation to methanol. They found the silica acts as powerful support and ligand to modify the cobalt species via a Co-O-SiOn linkage for turning the cobalt nanoparticles from catalysts for methanation/RWGS into catalysts for methanol production. The obtained results may benefit for further technology development of the production of methanol from CO₂ hydrogenation. In addition, the manuscript is well organized and clearly written. Some revisions are required before final acceptance by Nature Communications, as detailed in the following comments.

1. It was well known that cobalt catalysts have no active sites for WGS or RWGS. However, the CO selectivity over Co/SiO₂ or CoO_x (Traditional cobalt catalyst) is very high. How did the CO form over Co/SiO₂ or CoO_x?
2. The CO₂ conversion was very low (~8%) even at very high temperature (320 oC), which is not beneficial for the development of methanol synthesis from CO₂ hydrogenation. How to further enhance the catalytic activity?

3. From Fig. 2a, the methanol selectivity over Co/SiO₂ is much higher than that over CoOx at 320 oC, even is higher than that over Cu-based catalysts. However, the trace *CH₃O signal was found both on Co/SiO₂ and CoOx. Why? Maybe the temperature is high (350 oC) for In-situ IR measurement.

4. The CH₄ selectivity is very low over Co@Six catalysts. What is the mechanism for this phenomenon, low H₂ dissociation ability or high stability of *CH₃O intermediate species? The discussion on this point is not very clear.

5. The author found that the CO₂ conversion was not related to H₂ dissociation ability. Why did Co@Si_{0.95} exhibit higher CO₂ conversion? Which is the key factor?

Point-by-Point Responses to the Comments

Comments from Reviewer #1

This article proposed that the selectivity of CO₂ hydrogenation was dependent on the oxidation states of Co. Metallic Co caused the hydrogenation of CO₂ to CO. The oxidized Co in Co-O-Si center caused the hydrogenation of CO₂ to methanol. This is rather interesting. However, several major concerns still exist. This work cannot be accepted unless these issues are well addressed.

1-1. Comments: *There are obvious differences between in-situ XPS atmosphere treatment pressure conditions (pure H₂, 0.1 mbar; pure CO₂, 1.0 mbar; CO₂: H₂ = 3:1, 1:1 and 1:3, total pressure was 1.2 mbar, see SI in-situ XPS characterization for details) and the actual catalytic conditions (under reaction conditions of 1.0 to 4.0 MPa, see SI CO₂ hydrogenation for details). Can these in-situ XPS data reflect actual catalytic properties accurately under such different conditions? Under 1.0 mbar for in-situ XPS treatment, it is very likely that oxidized Co species are not completely reduced. However, under the actual catalytic conditions, the partial pressure of H₂ was much higher, so that Co might be completely reduced.*

Responses: Thanks for the insightful comments. Yes, the *in-situ* XPS atmosphere conditions are different from those in the practical catalytic tests, which are due to the limitation of the XPS technique. The XPS characterization must be performed under high-vacuum conditions for minimizing the loss of electrons in the path before detection (signals are originated from detecting the kinetic energy of the excited electrons on the solid surface) and avoiding the influence of pollutants on the solid surface. Currently, the extensively used and *state-of-the-art in-situ* XPS technique is to introduce the reactant gases with pressure at several mbar to the vacuum system to make balance between the treatment atmosphere and signal accuracy (*Catal. Sci. Technol.* **2019**, *9*, 3851–3867). Although such pressure of reactants is still lower than the practical reaction conditions, it is enough to react with the catalyst surface, detect the solid surface change and reaction intermediates by X-ray photoelectron signal, which has been widely used in various catalytic systems (*Science* **2014**, *345*, 546–550; *Nat. Energy* **2017**, *2*, 869–876; *Nat. Nanotech.* **2018**, *13*, 411–417; *J. Am. Chem. Soc.* **2016**, *138*, 13246–13252; *J. Am. Chem. Soc.* **2015**, *137*, 10104–10107; *Angew. Chem. Int. Ed.* **2016**, *55*, 7968–7973; *Chem* **2018**, *4*, 613–625; *ACS Catal.* **2019**, *9*, 4539–4550; *ACS Catal.* **2019**, *9*, 6783–6802; *Science* **2008**, *322*, 73–77; *Science* **2007**, *318*, 1757–1760; *Nat. Catal.* **2019**, *2*, 78–85; *Nat. Catal.* **2018**, *1*, 960–967). For example, Graciani *et al.* detected the carbon-containing intermediates on Cu/Ce catalyzed CO₂ hydrogenation using *in-situ* XPS with feed gas pressure at 0.0065 and 1 Torr (*Science* **2014**, *345*, 546–550).

In our work, in order to identify whether the employed conditions in *in-situ* XPS characterization could reduce the cobalt surface, we performed the *in-situ* XPS characterization of Co₃O₄/SiO₂ in 0.1 mbar of H₂. As shown in Reviewer-Only Figure 1, the sample was remarkably reduced, resulting the metallic cobalt at 500 °C, which is

in good agreement with the H₂-TPR measurement of cobalt oxide sample (Figure S29 in the revised SI). These data suggest that the employed reaction condition (0.1 mbar of H₂) is sufficient to reduce the cobalt oxides into metallic cobalt.

Under the equivalent treatment (0.1 mbar of H₂, 500 °C), the Co 2p XPS spectrum of Co₃O₄@Si_{0.95} still maintained the oxide surface as confirmed by the *in-situ* XPS spectra (Reviewer-Only Figure 2). Even after reduction at 600 and 650 °C, the sample gave the predominant Co²⁺ (781.5 eV) with some Co⁰ (777.8 eV). These data confirm that the cobalt species on Co@Si_{0.95} are more difficult to be reduced compared with the generally supported cobalt.

In order to provide more evidence, we treated the Co@Si_{0.95} sample with relatively high-pressure H₂ at 600 °C for 2 h (10% H₂ in Ar, 2.0 MPa), which should provide enough hydrogen for reducing the cobalt species. Interestingly, the treated Co@Si_{0.95} still exhibited cobalt oxide as a dominant phase with a small amount of metallic cobalt, as confirmed by the XRD (36.4°, 42.5° and 61.5° assigned to CoO phase) and XPS characterization (781.5 eV assigned to Co²⁺ and 777.8 eV assigned to Co⁰, Reviewer-Only Figure 3). Even after reaction for 100 h in the practical CO₂ hydrogenation conditions, the Co@Si_{0.95} sample still exhibited a major CoO phase with a relatively small amount of metallic Co (see XRD and XPS in Reviewer-Only Figure 4), confirming the difficulty for reduction of cobalt species on the Co@Si_{0.95} catalyst, in good agreement with those in the *in-situ* XPS investigation.

In summary, the gaseous hydrogen is sufficient to reduce the cobalt catalyst in the *in-situ* XPS tests. In the revised manuscript, we employed not only the *in-situ* XPS but also the practical reduction and catalysis treatment to the samples for studying the change in cobalt oxidation state. We have added these data and discussion in the revised manuscript.

1-2. Comments: *The published literatures show that Co⁰ tends to show high methane selectivity in CO₂ hydrogenation (Journal of CO₂ Utilization, 2018, 26: 221-229). In this article, Co/SiO₂ was synthesized as the comparison sample. A series of characterizations shows that Co in Co/SiO₂ was basically at metallic states. However, in this article, the major product for Co/SiO₂ was CO (see Figure 2a). Why did the selectivity of metallic Co in this work differ from those previously reported?*

Responses: Thanks for the comments. Yes, we highly agree that methane is usually formed on the cobalt catalysts in CO₂ hydrogenation, but the methane and CO selectivities depend on the multiple factors (e.g. CO₂ conversion, reaction temperature, pressure, catalyst amount, and the support, *ChemCatChem* **2019**, *11*, 4884–4893; *Catal. Sci. Technol.* **2016**, *6*, 7449–7460; *Catal. Today* **2019**, *337*, 162–170; *ACS Catal.* **2019**, *9*, 2739–2751; *Appl. Catal. B* **2018**, *220*, 397–408; *J. CO₂ Util.* **2018**, *26*, 221–229; *ACS Catal.* **2016**, *6*, 2811–2818; *J. Am. Chem. Soc.* **2017**, *139*, 9739–9754; *Angew. Chem. Int. Ed.* **2014**, *53*, 6705–6709; *Angew. Chem. Int. Ed.* **2016**, *55*, 7968–7973; *Ind. Eng. Chem. Res.* **2013**, *52*, 2247–2256). In our experiments, we found that the Co/SiO₂ and CoO_x samples with metallic cobalt species exhibited higher methane selectivity than the Co@Si_x catalyst in CO₂ hydrogenation. In these cases, the methane selectivity

is still slightly lower than CO (Figure 2a in the main text), which should be due to the employed reaction conditions, and similar phenomena have been observed previously (*ChemCatChem* **2019**, *11*, 4884–4893; *ACS Catal.* **2019**, *9*, 2739–2751; *ACS Catal.* **2016**, *6*, 2811–2818; *J. Am. Chem. Soc.* **2017**, *139*, 9739–9754).

We explored the methane selectivity as a function of reaction temperature in the Co/SiO₂ and CoO_x catalyzed CO₂ hydrogenation. As shown in Reviewer-Only Figures 5a and 6a, the methane selectivity remarkably increased with the reaction temperatures. For example, the Co/SiO₂ gave CH₄ selectivity at 24.7% at 300 °C. By increasing the reaction temperature to 380 °C, the CH₄ selectivity raised to 54.9%. For the CoO_x catalyst, the CH₄ selectivity was 35.4% at 300 °C and then increased to 73.2% at 380 °C. Further increasing the reaction temperature to 500 °C gave lower CH₄ selectivity at 52.2% (Reviewer-Only Figures 7a), which might be attributed to the strong exothermic nature of the CO₂ methanation, leading to thermodynamically favorable CO formation *via* reverse water–gas shift reaction and decreased CH₄ selectivity (*J. CO₂ Util.* **2017**, *17*, 312–319). These data suggest that the reaction temperatures have a significant effect on the CH₄ and CO selectivity (*ChemCatChem* **2019**, *11*, 4884–4893; *Catal. Sci. Technol.* **2016**, *6*, 7449–7460; *ACS Catal.* **2016**, *6*, 2811–2818).

In addition to the reaction temperature, we also explored the influences of reaction pressure, feed gas composition, and catalyst amount to the methane selectivity in CO₂ hydrogenation (Reviewer-Only Figures 7b–d). Higher reaction pressure, more hydrogen feed, and larger catalyst amount (low GHSV) are regarded to be the factors for enhancing the methane selectivity.

We found that different supports for cobalt species also significantly influenced the CH₄ selectivity in CO₂ hydrogenation. Reviewer-Only Figure 8 shows the data characterizing the performance of Co nanoparticles (20 wt%) supported on SiO₂, TiO₂ (anatase and rutile), Al₂O₃ (denoted as 20Co/SiO₂, 20Co/TiO₂ and 20Co/Al₂O₃, respectively) in CO₂ hydrogenation. The 20Co/Al₂O₃ and 20Co/TiO₂(rutile) exhibited much higher methane selectivity than other catalysts. For example, the 20Co/TiO₂(rutile) gave the highest CH₄ selectivity of 76.3% at 400 °C, while 20Co/Al₂O₃ showed CH₄ selectivity at 58.1%. The silica supported catalysts (20Co/SiO₂ and Co/SiO₂) showed medium CH₄ selectivity at 42.5–45.1%. Notably, the 20Co/TiO₂(anatase) exhibited a really low CH₄ selectivity of 26.5%. This phenomenon is in good agreement with that in the previous works (*ACS Catal.* **2019**, *9*, 2739–2751; *Angew. Chem. Int. Ed.* **2014**, *53*, 6705–6709; *J. Am. Chem. Soc.* **2017**, *139*, 9739–9754), which might be attributed to the distinguishable interaction between the cobalt and the oxide support.

On the basis of the aforementioned investigations, we confirm that Co-based catalysts are active for CO₂ methanation. Their catalytic performance, for instance, CO₂ conversion and CH₄ selectivity, are strongly influenced by many factors. Therefore, under the given reaction conditions in Figure 2a, the methane selectivity of Co/SiO₂ and CoO_x are reasonable. We have partially added these results and discussion in the revised manuscript.

1-3. Comments: *Metallic Co is a conventional FTS active site which can catalyze*

the hydrogenation of CO to long chain products. For Co/SiO₂, why were the produced CO molecules not hydrogenated into C₂₊ products?

Responses: Thanks for the insightful comments. Yes, we agree that Co-based catalysts are widely used in Fischer–Tropsch synthesis for long chain products. Our experiments also indicate that the cobalt catalysts are active for the C–C couplings. Reviewer-Only Figures 5 and 6 show the CO₂ conversion and product selectivity over Co/SiO₂ and CoO_x catalyst, respectively. The C₂₊ hydrocarbons are detectable with selectivity of 4.0%–8.5% on Co/SiO₂ and 2.3%–12.4% on CoO_x at 260–380 °C. The lower C₂₊ selectivities in this process than the general Fischer–Tropsch process might be due to abundant CO₂ and scarce CO in the mixed gas. We have added these data and discussion in the revised manuscript.

1-4. Comments: In terms of reaction mechanism, why can the stabilization of CH₃O improve methanol selectivity? In my opinion, higher methanol selectivity and lower CO selectivity are mainly due to HCOO* intermediate species. According to the review (Nanoscale, 2015, 7, 8663), HCOO* pathway does not produce CO. In addition, if the adsorption of CH₃O* is strong, will methanol be over hydrogenated to CH₄? TPSR-MS experiment is suggested to find out from what temperature methanol will be hydrogenated to methane.*

Responses: Thanks for the comments. We made detailed responses to the questions as follows:

(i)HCOO pathway does not produce CO.....*

It is recognized that there are two major reaction pathways for the CO₂-to-CH₃OH transformation based on experimental observation and theoretical calculations, including the RWGS + CO hydrogenation pathway and the formate pathway (*Science* **2017**, 355, 1296–1299; *Science* **2012**, 336, 893–897; *Science* **2014**, 345, 546–550; *J. Am. Chem. Soc.* **2017**, 139, 9739–9754; *J. Am. Chem. Soc.* **2016**, 138, 12440–12450; *J. Am. Chem. Soc.* **2015**, 137, 8676–8679; *Nat. Chem.* **2014**, 6, 320–324; *J. Catal.* **2009**, 263, 114–122; *J. Catal.* **2014**, 317, 44–53; *ACS Catal.* **2013**, 3, 1296–1306; *ACS Catal.* **2011**, 1, 365–384; *ChemCatChem* **2015**, 7, 1105–1111; *J. Catal.* **2011**, 281, 199–211). In the RWGS + CO hydrogenation pathway, the *CO intermediate is firstly produced from the RWGS reaction *via* the *HOCO intermediate, or the direct C–O bond cleavage of *CO₂. The *CO intermediate is further hydrogenated to *CH₃O, which could be transformed into CH₃OH. Simultaneously, the *CO species could also desorb from the catalyst directly to form CO, which have been reported on the Co-based catalysts (*ChemCatChem* **2019**, 11, 4884–4893; *Angew. Chem. Int. Ed.* **2014**, 53, 6705–6709; *ACS Catal.* **2019**, 9, 2739–2751; *Catal. Sci. Technol.* **2016**, 6, 7449–7460; *ACS Catal.* **2016**, 6, 2811–2818). The formate pathway proceeds the *HCOO species by the primary hydrogenation of CO₂, which produces *CH₃O *via* the C–O bond cleavage of the *H_xCOOH intermediates, and is eventually hydrogenated to the CH₃OH. According to such proposed pathway, it seems the reaction does not produce CO (*J. Am. Chem.*

Soc. **2017**, *139*, 9739–9754). However, we cannot exclude the CO production from the formate route because the formic acid is easily decomposed into CO and water on the metal catalysts (*Angew. Chem. Int. Ed.* **2013**, *52*, 4406–4409; *J. Phys. Chem. C* **2018**, *122*, 20279–20288; *ChemSusChem* **2015**, *8*, 260–263). In addition, the *CH₃O and methanol species could be decomposed into CO species.

(ii) *why can the stabilization of *CH₃O improve methanol selectivity*

Notably, the *CH₃O is a key intermediate in both RWGS + CO hydrogenation and formate routes, the selective hydrogenation of *CH₃O into CH₃OH is crucial for the methanol production (*Science* **2017**, *355*, 1296–1299; *Science* **2012**, *336*, 893–897; *Science* **2014**, *345*, 546–550; *J. Am. Chem. Soc.* **2017**, *139*, 9739–9754; *J. Am. Chem. Soc.* **2015**, *137*, 10104–10107; *J. Am. Chem. Soc.* **2016**, *138*, 12440–12450; *Angew. Chem. Int. Ed.* **2017**, *56*, 1–7; *J. Am. Chem. Soc.* **2015**, *137*, 8676–8679; *Sci. Adv.* **2017**, *3*, e1701290). However, according to the previous theoretical simulations, the *CH₃O hydrogenation to *CH₃OH proceeds high energy barriers ($E_a = 1.49$ eV). The easily detected abundant *CH₃O signals in the *in-situ* DRIFT and XPS characterization also confirm the fast formation and slow further transformation of *CH₃O on the Co@Si_x catalysts. Apart from the hydrogenation to *CH₃OH, the *CH₃O species might also undergo the C–O cleavage and the subsequent hydrogenation to CH₄ (*J. Am. Chem. Soc.* **2017**, *139*, 9739–9754; *Science* **2017**, *355*, 1296–1299; *J. Am. Chem. Soc.* **2016**, *138*, 12440–12450), as well as the dehydrogenation to CO. Therefore, the stabilization of *CH₃O species on the catalyst surface to avoid the C–O cleavage and deep dehydrogenation could improve the methanol selectivity.

(iii) *...will methanol be over hydrogenated to CH₄? ...TPSR-MS experiment...*

Yes, we agree that the methanol might be over hydrogenated to CH₄ on the supported metal catalysts *via* the C–O cleavage (*J. Am. Chem. Soc.* **2017**, *139*, 9739–9754; *Angew. Chem. Int. Ed.* **2016**, *55*, 7968–7973; *J. Catal.* **2016**, *343*, 115–126; *J. Am. Chem. Soc.* **2015**, *137*, 8676–8679). Following the reviewer's suggestion, we performed the TPSR experiments with methanol as a feed. Because methanol easily forms the *CH₃O species, such experiment also reveals the transformation of *CH₃O as a function of the temperature (*Angew. Chem. Int. Ed.* **2019**, *58*, 11242–11247; *J. Am. Chem. Soc.* **2017**, *139*, 9739–9754; *Angew. Chem. Int. Ed.* **2016**, *55*, 7968–7973; *J. Catal.* **2016**, *343*, 115–126). As shown in Reviewer-Only Figure 9, the CO signal (*m/z* at 28) appeared on both samples, demonstrating the dehydrogenation of the *CH₃O species indeed occurred on the surface of metallic cobalt. In addition, the methane signal was detected at 340 °C on Co/SiO₂ catalyst, assigned to the C–O dissociation and deep hydrogenation occurred at such temperature. In contrast, no methane signal was observed on Co@Si_{0.95} catalyst, even at temperature up to 420 °C. These results confirm the hindered C–O cleavage on the Co@Si_{0.95} catalyst, which stabilizes the *CH₃O intermediate to benefit the methanol production rather than methanation.

The previous DFT calculations have revealed that the cleavage of C–O bond on *CH₃O requires the metallic Co surface or the CoO surface with abundant oxygen vacancies. For example, the CoO(100) surface with saturated oxygen leads to high

energy barrier for the $^*\text{CH}_3\text{O}$ dissociation at 2.71 eV [1.45 eV for Co(111) surface and 1.01 eV for the oxygen vacancy-rich CoO(100)]. The Co@Si_{0.95} catalyst with not-easy-to-reduce oxygen species provided ideal catalyst surface for hindering the C–O cleavage, which accelerates the $^*\text{CH}_3\text{O}$ hydrogenation to methanol rather than C–O cleavage to form methane. Apart from the $^*\text{CH}_3\text{O}$ decomposition, another possible route for methane formation proceeds the direct $^*\text{CO}$ dissociation into $^*\text{C}$ species, which has been experimentally and theoretically studied in the cobalt-catalyzed Fischer–Tropsch synthesis. The metallic cobalt and cobalt carbide are found to be efficient for CO dissociation, but the oxidized cobalt surface is known to be less active, which is also confirmed by the poor activity of Co@Si_{0.95} in the CO hydrogenation (CO conversion of 0.7% and methanol selectivity of 22.7%) under the employed reaction conditions (360 °C, 2.0 MPa, Figure S51 in the revised SI).

In summary, all these data confirm that the Co@Si_x catalysts could stabilize the $^*\text{CH}_3\text{O}$ species to minimize the side reactions, such as the C–O cleavage and hydrogenation to form methane, dehydrogenation to form CO. Owing to these features, the Co@Si_x catalysts exhibit high selectivity for the methanol formation in the multiple competitive reaction pathways.

We have added these data and discussion in the revised manuscript.

1-5. Comments: *In this article, the authors only provided XRD data of different catalysts before and after reduction treatment. The comparison of catalysts structure and phase composition before and after reaction is not included. Then, will Co in Co@Si_x be partially carbonized after a long run?*

Responses: Thanks for the comments. Yes, we agree that the cobalt species might be transformed into cobalt carbide in the CO/CO₂ hydrogenation system (*ACS Catal.* **2016**, *6*, 913–927; *ACS Catal.* **2019**, *9*, 9554–9567; *ACS Catal.* **2018**, *8*, 228–241; *ACS Catal.* **2017**, *7*, 8285–8295; *ACS Catal.* **2017**, *7*, 8023–8032; *ACS Catal.* **2015**, *5*, 3620–3624). Following the reviewer’s suggestion, we characterized the Co@Si_{0.95} and Co/SiO₂ catalysts after the catalysis in CO₂ hydrogenation for 100 h. As shown in Reviewer-Only Figure 10, the used Co/SiO₂ presented dominant metallic Co accompanied with Co₂C species. The HRTEM images (Reviewer-Only Figure 11) also confirm the coexistence of Co and Co₂C phases. These data confirm that the cobalt species on Co/SiO₂ are partially carbonized during the catalysis, which is in good agreement with the known phenomena. Interestingly, after reaction for 100 h under the equivalent conditions, the used Co@Si_{0.95} still exhibited dominant CoO phase accompanied by metallic Co phase with undetectable Co₂C/CoC species, as confirmed by the XRD pattern and HRTEM images (Reviewer-Only Figures 4 and 12). The remarkably different phenomena on Co@Si_{0.95} compared with the general cobalt catalysts should be due to the Co–O–Si linkage on the Co@Si_{0.95} catalyst, which hindered the carbonization (*J. Am. Chem. Soc.* **2012**, *134*, 15814–15821; *J. Phys. Chem. C* **2017**, *121*, 5154–5160; *Nat. Commun.* **2014**, *5*, 5783). We have added these data and discussion in the revised manuscript.

1-6. Comments: In my opinion, when comparing the performance of different catalysts, the optimal conditions for each sample should be chosen. For Cu/ZnO/Al₂O₃, it is generally considered that the optimal operation temperature should be lower than 300 °C (Science, 2012, 336, 893-897). In this paper, the performance was compared at 320 °C, which was higher than the optimum temperature of Cu/ZnO/Al₂O₃. In addition, the performance of Cu/ZnO/Al₂O₃ used in this paper was too poor, as the conversion was only 1% at 200 °C, and methanol selectivity was just 60%. Compared with the data of Cu/ZnO/Al₂O₃ (conversion= \sim 8%, methanol selectivity= \sim 80%, at 200 °C) in Science Advances, 2017, 3: e1701290. The authors can directly use commercial Cu/ZnO/Al₂O₃ as comparison.

Responses: Thanks for the comments. We agree that the catalysis on Cu/ZnO/Al₂O₃ should be performed on the optimal temperatures. Accordingly, we performed the catalysis under optimized temperatures using commercial Cu/ZnO/Al₂O₃ catalyst. Reviewer-Only Figure 13 shows the data characterizing the performance of commercial Cu/ZnO/Al₂O₃ catalyst in a wide temperature range (200–380 °C). The methanol selectivity of 82.3% appeared at 200 °C with CO₂ conversion of 7.6%, and then the CO₂ conversion increased and methanol selectivity decreased with raising the reaction temperatures. The best CH₃OH yield appeared at 240 °C, giving 70.8 mmol g_{cat}⁻¹ h⁻¹, which is higher than that of Co@Si_{0.95} catalyst (59.7 mmol g_{cat}⁻¹ h⁻¹). However, the Cu/ZnO/Al₂O₃ suffers from poor durability, giving remarkably reduced performance in the reaction life test. For example, after reaction for 50 h at 240 °C, almost half of the methanol yields was lost on the Cu/ZnO/Al₂O₃ catalyst (Reviewer-Only Figure 14). This phenomenon is in agreement with the knowledge of Cu/ZnO/Al₂O₃ catalyst, where the Cu nanoparticles easily sintered into larger ones and caused the deactivation (Nat. Mater. 2013, 12, 34–39; Angew. Chem. Int. Ed. 2016, 55, 12708–12712; Nat. Commun. 2013, 4, 2339; ACS Catal. 2015, 5, 4439–4448; ACS Catal. 2017, 7, 912–918). In contrast, the Co@Si_{0.95} catalyst has superior durability at even higher temperature, exhibiting almost unchanged performance before and after the tests for 100 h (70 h at 320 °C and 30 h at 380 °C, Figure 2d in the main text). We have added these data and discussion in the revised manuscript.

1-7. Comments: In the synthesis process of Co@Si_x, no surface treatment (such as adding TCAC as coating agent) was carried out on the particles obtained by coprecipitation to control the uniform coating of precipitated particles. Theoretically, the catalyst obtained by this method contained both Co coated by SiO_n and Co loaded on the surface. There should be differences in catalytic activity between these two Co sites. I suggest the authors to provide further evidences to clarify the distribution of Co centers in Co@Si_x and the contribution of different centers to the overall catalytic activity.

Responses: Thanks for the valuable comments. Yes, the Co@Si_x catalysts were directly obtained by coprecipitation of the cobalt and silica precursors under the basic conditions. In this process, the cobalt phyllosilicates were easily formed with abundant

Co–O–SiO_x linkages, as confirmed by the characteristic peaks in IR spectra (Figure 3i in the main text) and typical lamellar structure observed in the TEM image (Figure 3a in the main text). After reduction with hydrogen, part of the cobalt species could be reduced into metallic cobalt and the cobalt species interacting with silica still maintained the oxidation state because of the Co–O–SiO_x interaction. According to our characterization data and the previous investigations on the phyllosilicates-derived catalysts (*J. Am. Chem. Soc.* **2012**, *134*, 13922–13925; *J. Catal.* **2013**, *297*, 142–150; *Nat. Commun.* **2013**, *4*, 2339; *Nat. Commun.* **2018**, *9*, 3367), the Co@Si_x catalysts should have two types of cobalt species, including the metallic cobalt on the cobalt surface and CoO on the cobalt–silica interface. No matter the cobalt nanoparticles on the surface or within the amorphous sheath, the Co interacting with silica should be Co–O–SiO_x and the bare surface should be Co⁰. The scheme showing the catalyst structure is given in Reviewer-Only Scheme 1.

The M–O–SiO_x interface has been widely investigated in hydrogenation reactions. Although the role of such interface has not been fully understood, the synergy of the metallic phase and oxide phase are regarded to be crucial (*J. Am. Chem. Soc.* **2012**, *134*, 13922–13925; *J. Catal.* **2013**, *297*, 142–150; *Nat. Commun.* **2013**, *4*, 2339; *Nat. Commun.* **2018**, *9*, 3367). Generally, the metal catalyzed hydrogenation processes involve in the steps including (i) dissociation of hydrogen, (ii) adsorption of unsaturated compounds, (iii) stepwise hydrogenation with H atoms. According to such knowledge, the metallic phase (e.g. Co⁰) might activate H₂ and the oxide phase (e.g. Co^{δ+} species) adsorbs CO₂. This proposed pathway is in good agreement with the general knowledge on the Cu–O–SiO_x catalysts (*J. Am. Chem. Soc.* **2012**, *134*, 13922–13925; *J. Catal.* **2013**, *297*, 142–150; *Nat. Commun.* **2013**, *4*, 2339). In addition, the previous study has revealed that the Cu–O(H)–SiO_x interface accelerate the hydrogen activation/splitting (*Nat. Commun.* **2018**, *9*, 3367). In order to identify whether the Co@Si_x catalysts proceeds similar mechanism, we removed partial silanol groups by treating the sample with NaOH. The resulted Co@Si_{0.95}-Na catalyst exhibited remarkably reduced hydrogenation activity than the untreated Co@Si_{0.95} (Reviewer-Only Figure 15), suggesting the important role of Co–O–SiO_x interface for the reaction.

In sum, both the metallic Co⁰ and interfacial Co^{δ+}–O–SiO_x might contribute to the hydrogen activation and accelerate the hydrogenation reactions. The interfacial Co^{δ+}–O–SiO_x species stabilized the crucial reaction intermediate (e.g. *CH₃O) and improved the methanol selectivity, while the metallic Co⁰ would catalyze the *CH₃O decomposition to form CO and methanol. By optimizing the composition, the best catalyst was realized as Co@Si_{0.95}. We have partially added these data and discussion in the revised manuscript.

1-8. Comments: *The electron micrographs and a series of spectral characterizations provided in this article did not reflect the difference between Co centers in Co@Si_x with different x values. However, when x changed from 0.52 to 1.58, the selectivity of Co@Si_x changed significantly (Figure 2a). Why was the selectivity dependent on the x value?*

Responses: Thanks for the comments. Yes, the catalytic performances are different on the various Co@Si_x samples, which is assigned to that the different Co/Si ratios strongly influenced the oxidation state of cobalt species. In the HRTEM images, both metallic cobalt and CoO phases are observed on the Co@Si_{0.52} and Co@Si_{0.96} samples. When the silica amount is increased, the Co@Si_{1.48} and Co@Si_{1.87} samples exhibited CoO phases with almost undetectable metallic cobalt (Figures S14–S18 in the revised SI).

Reviewer-Only Figure 16 shows the H₂-TPR profiles of these samples. The Co₃O₄ without silica species showed the reduction peaks at 387 and 494 °C, which are assigned to the reduction of Co³⁺ to Co²⁺ and Co²⁺ to Co⁰. When the cobalt oxide was modified by silica, these peaks were remarkably shifted to higher temperature and the Co²⁺ to Co⁰ signal was weakened. For example, the Co₃O₄@Si_{0.52} showed the Co³⁺ to Co²⁺ signal at 402 °C and weak Co²⁺ to Co⁰ signal at 533 °C. The sample with lower Co/Si ratio, such as the Co₃O₄@Si_{0.95} sample, exhibited even higher Co³⁺ to Co²⁺ signal at 410 °C and almost undetectable Co²⁺ to Co⁰ signal. These data demonstrate the different silica amount strongly influences the oxidation state of cobalt nanoparticles, which is further confirmed by the Co 2p XPS (Figure S24 in the revised SI) and Raman spectra (Figure S31 in the revised SI).

As shown in Figure 2a, the methane selectivity on the Co@Si_{0.52} is higher than the other Co@Si_x with more silica species, which is due to the different metallic cobalt and cobalt oxide content on these samples. The Co@Si_{0.52} with more metallic Co benefits the methanation. With regard to the CO and methanol selectivity, the different amount of silica–cobalt interaction might influence the stability of *CH₃O species because it could form methanol or CO by hydrogenation or dehydrogenation, respectively. By optimizing the Co/Si ratios, the best methanol selectivity was achieved on Co@Si_{0.95}. Apart from the cobalt oxidation state, more silica species might block more surface sites of the Co@Si_x catalysts, which also influences the catalytic performance. We have added these data and discussion in the revised manuscript.

1-9. Comments: *According to the description in the part of SI In-situ FT-IR characterization (IR spectra were recorded using a Thermo Fisher Nicolet iS50 FT-IR spectrometer equipped with a MCT / a detector), the author used the Drifts mode instead of FT-IR mode when collecting infrared data. The ordinate in the infrared spectrum of this article is marked incorrectly. For the Drifts spectrum, the ordinate is not "absorption". In addition, in this part of the description of the infrared test experiment, the pressure data during the test was not clearly provided. Were there any differences between the experiment condition and catalytic conditions?*

Responses: Thanks for the insightful comments. The ordinate should be Kubelka-Munk in the infrared spectra conducted in Drifts mode (Figure 4a in the main text and Figures S40–S43 in the revised SI). The *in-situ* Drifts characterization was performed at under ambient pressure and different from that in the practical catalytic tests, but it can reveal the interaction between the reactant molecules with the catalyst surface, and similar strategy has been extensively used for studying the reaction mechanisms

(*Science* **2014**, *345*, 546–550; *Nat. Chem.* **2017**, *9*, 120–127; *Nat. Energy* **2017**, *2*, 869–876; *Nat. Nanotech.* **2018**, *13*, 411–417; *Nat. Commun.* **2019**, *10*, 1166; *Nat. Commun.* **2018**, *9*, 3457; *Sci. Adv.* **2017**, *3*, e1701290; *J. Am. Chem. Soc.* **2017**, *139*, 6827–6830; *J. Am. Chem. Soc.* **2019**, *141*, 8482–8488; *J. Am. Chem. Soc.* **2016**, *138*, 6298–6305; *J. Am. Chem. Soc.* **2016**, *138*, 12440–12450; *Angew. Chem. Int. Ed.* **2018**, *57*, 6104–6108; *Chem* **2018**, *4*, 613–625; *Joule* **2019**, *3*, 570–583). We have added the information and changed the ordinate in the Drifts data in the revised manuscript.

1-10. Comment: For revealing the reaction path, I suggest the authors to conduct pressure-dependent tests to figure out the reaction orders.

Responses: Thanks for the comments. We performed pressure-dependent tests and showed the reaction orders of CO₂ and H₂. As shown in Reviewer-Only Figure 17, an approximately first-order dependence on the H₂ partial pressure is observed for methanol synthesis, giving 0.80 and 0.91 over Co@Si_{0.95} and Co/SiO₂ catalysts, respectively. This result indicates that the H₂ positively influences the methanol formation in the CO₂ hydrogenation (*ACS Catal.* **2011**, *1*, 365–384; *Catal. Sci. Technol.* **2017**, *7*, 3375–3387; *ACS Catal.* **2019**, *9*, 8785–8797; *J. Catal.* **2019**, *369*, 415–426; *J. Catal.* **1995**, *156*, 229–242; *J. Catal.* **2015**, *328*, 43–48; *J. Catal.* **2013**, *300*, 141–151; *J. Catal.* **2013**, *298*, 10–17; *Catal. Today* **2016**, *270*, 31–42). The apparent CO₂ reaction order in methanol synthesis is close to zero (0.09 for Co@Si_{0.95} and 0.11 for Co/SiO₂), which should be due to that the catalyst surface is saturated by CO₂ and the reaction intermediates, and higher hydrogen pressure could efficiently accelerate the whole reactions.

Although the reaction pathways for CO₂ hydrogenation have not been fully understood, it is generally recognized that there are two major reaction pathways for the CO₂-to-CH₃OH transformation based on experimental observation and theoretical calculations, including the RWGS + CO hydrogenation pathway and the formate pathway (*Science* **2017**, *355*, 1296–1299; *Science* **2012**, *336*, 893–897; *Science* **2014**, *345*, 546–550; *J. Am. Chem. Soc.* **2017**, *139*, 9739–9754; *J. Am. Chem. Soc.* **2016**, *138*, 12440–12450; *J. Am. Chem. Soc.* **2015**, *137*, 8676–8679; *Nat. Chem.* **2014**, *6*, 320–324; *J. Catal.* **2009**, *263*, 114–122; *J. Catal.* **2014**, *317*, 44–53; *ACS Catal.* **2013**, *3*, 1296–1306; *ACS Catal.* **2011**, *1*, 365–384; *ChemCatChem* **2015**, *7*, 1105–1111; *J. Catal.* **2011**, *281*, 199–211). Different rate-determine steps have been suggested for methanol pathway from CO₂ and H₂ via these pathways. For example, in the formate pathway, hydrogenation of *HCOO (*HCOO + H → *HCOOH) and *CH₃O (*CH₃O + H → *CH₃OH) intermediates are the rate-control steps (*J. Am. Chem. Soc.* **2017**, *139*, 9739–9754; *Science* **2017**, *355*, 1296–1299; *J. Am. Chem. Soc.* **2016**, *138*, 12440–12450). As reported previously, if the hydrogenation of *HCOO is the rate-control step, the maximum attainable reaction orders according are 1 for both CO₂ and H₂. Generally, H* and HCOO* are always abundant on the catalyst surface (*ACS Catal.* **2011**, *1*, 365–384; *Catal. Sci. Technol.* **2017**, *7*, 3375–3387; *J. Catal.* **1995**, *156*, 229–242), the apparent CO₂ and H₂ reaction orders would be much smaller than 1, which is inconsistent with our results (H₂ reaction order is close to 1). These data confirm that

the H₂ or CO₂ activation should not be the control step. Our *in-situ* DRIFT and XPS results demonstrate that the *CH₃O was abundantly detected on the catalyst surface in CO₂ and H₂ atmosphere, and the further transformation of *CH₃O was slow, suggesting the *CH₃O hydrogenation should be the rate-control step. The low reaction order of CO₂ (0.09–0.11) can be attributed to the slow step of *CH₃O hydrogenation to CH₃OH (*ACS Catal.* **2011**, *1*, 365–384; *Catal. Sci. Technol.* **2017**, *7*, 3375–3387). Similarly, if CO₂ hydrogenation follows the RWGS + CO hydrogenation pathway, the *CH₃O is also the crucial intermediate for CH₃OH formation, and *CH₃O hydrogenation to CH₃OH is the rate-control step (*J. Am. Chem. Soc.* **2017**, *139*, 9739–9754; *Science* **2017**, *355*, 1296–1299; *J. Am. Chem. Soc.* **2016**, *138*, 12440–12450).

These kinetic data demonstrate that methanol production over the Co@Si_{0.95} and Co/SiO₂ catalysts follows similar route that *CH₃O hydrogenation to CH₃OH is the rate-control step. The Co@Si_{0.95} benefits this step and accelerates the methanol production, in good agreement with those in *in-situ* DRIFTS, XPS, and TPSR investigations. We have partially added these data and discussion in the revised manuscript.

Comments from Reviewer #2

This work is concerned with the conversion of CO₂ to methanol over a silica modulated cobalt catalysts. The novelty of this work is the development of a stable, highly active methanol synthesis cobalt catalysts via the formation of Co phyllosilicates intermediates. This work used a judicious number of characterization techniques, both in situ and ex situ, to support the structure/activity claims. This work can potentially help to pave the way for the use of modulated support species on other transition metals to control the extent of hydrogenation. Overall, I could recommend this work for publication with major revisions addressing the following concerns.

2-1. Comments: *While the Co₃O₄@Si_x and Co@Si_x were rigorously characterized, there was no characterization of the base SiO_x material using the same preparation method. Furthermore, the surface areas of these catalysts were not explored. The surface area and overall porosity of the different Co@Si_x could have a considerable influence on the performance, where the influence of mass transfer was only briefly discussed for the high Co/Si samples.*

Responses: Thanks for the comments. Following the reviewer's suggestion, we synthesized bare SiO_x material without cobalt species by the hydrolysis of TEOS with the existence of NH₃·H₂O, showing the XRD patterns assigned to typical amorphous silica (Reviewer-Only Figure 18).

In addition, we also agree that the surface area of the silica-containing catalysts should be provided. As shown in Reviewer-Only Table 1, the Co@Si_x and Co/SiO₂ catalysts showed the surface areas at 123.5–157.6 m²/g and mesoporous volumes at 0.17–0.26 cm³/g. Similar mesoporosity in these samples means the comparable mass transfer, which should not be the reason for the different catalytic performances.

We have added these data in the revised manuscript.

2-2. Comments: *The XANES spectra shown in Figure 2j should be normalized to an edge step of 1 such that comparisons between the sample can be readily accessed. An inset of the pre-edge region should be provided in the supplemental information that illustrates the differences in the pre edge regime to help guide the reader.*

Responses: Thanks for the comments. We have normalized the XANES spectra in Figure 3j to the edge step of 1 for clear comparison (Reviewer-Only Figure 19), and the enlarged pre-edge region is also provided as an inset of the figure.

2-3. Comments: *The EXAFS spectra in Figure 2k was fitted using single scattering paths from Co-O and Co-Co (from a CoO reference) and a Co-Co (from a metallic Co reference) which gave rise to reasonable radius for the specified paths. However, the authors should compare the direct XANES measurements with the XPS of the samples before in situ reductive pretreatment and determine if the result are in agreement. The comparison of the Co 2p spectra for all of the Co@Si_x samples should be added in a compiled supplemental figure so that the reader can understand the starting surface composition of the catalysts before exposure to hydrogen or reaction conditions.*

Responses: Thanks for the comments. Yes, we agree that the surface composition of the starting materials (Co₃O₄@Si_x and Co₃O₄/SiO₂) before H₂ reduction and CO₂ hydrogenation are important. Following the reviewer's suggestion, we performed XANES and XPS characterization of these samples. As shown in Reviewer-Only Figure 20, all of Co₃O₄@Si_x and Co₃O₄/SiO₂ samples exhibit a pre-edge feature at 7710.4 eV and the first-order derivative peaks at 7717.5 eV and 7721.5 eV, which are similar to the pre-edge features of Co₃O₄ and the edge positions of Co²⁺ and Co³⁺ cations, respectively (*Appl. Catal. A* **2006**, 312, 12–19; *Energy Environ. Sci.* **2013**, 6, 926–934). Reviewer-Only Figure 21 shows the Co 2p XPS spectra of Co₃O₄ and Co₃O₄@Si_x samples, giving obvious peaks assigned to Co²⁺/Co³⁺ species (Reviewer-Only Figures 3 and 4), in agreement with those in XANES and order characterization. We have added the related results and discussions in our revised manuscript.

2-4. Comments: *The authors observed noticeably different reaction intermediates on the Co@Si_{0.95} catalyst relative to the control Co/SiO₂ sample, noticeably the formation of the *CH₃O intermediate which was verified via both in situ DRIFTS and XPS. The argument for the difference in reaction mechanism would be strengthened via the inclusion of apparent activation energies for the catalysts.*

Responses: Thanks for the comments. According to the reviewer's suggestion, we performed the kinetic experiments and present the apparent activation energies (E_a) for CO, CH₃OH and CH₄ formation over the Co@Si_{0.95} and Co/SiO₂ catalysts. As shown in Reviewer-Only Figure 22, the apparent E_a for methane and CO production on Co@Si_{0.95} are 135.5 and 58.2 kJ mol⁻¹, respectively. These values are remarkably higher than those over the Co/SiO₂ catalyst (96.4 kJ mol⁻¹ for methane and 43.1 kJ mol⁻¹ for CO production). These data confirm easier methane and CO formation on the Co/SiO₂

catalyst than that on the Co@Si_{0.95}. In addition, the Co@Si_{0.95} exhibited apparent E_a for methanol production at 58.2 kJ mol⁻¹, which is lower than 62.4 kJ mol⁻¹ on the Co/SiO₂ catalyst, suggesting that easier methanol production on Co@Si_{0.95} catalyst than that on the Co/SiO₂ catalyst. We have added these data and discussion in the revised manuscript.

2-5. Comments: *The authors attribute the increase in catalytic performance to the formation of Co-O-SiO_n linkages that serve to stabilize the active CoO species, however, all of the Co@Si_x catalysts have constant particle size of ~3.8 nm. Given that the cobalt particle sizes are uniform within error, it would be reasonable to assume that the extent of interfacial Co-O-Si linkages should be constant throughout the catalyst. Therefore, the authors should give a more detailed description of the influence of the Co/Si ratio on the active site, where the increase in Co/Si ratio should result in higher mass transfer limitations.*

Responses: Thanks for the comments. Yes, we agree that the catalysts with different Co/Si ratio have similar cobalt nanoparticle diameters but significantly different catalytic performance, which should be due to the distinguishable cobalt oxidation state rather than the mass transfer. The detailed explanation is in the following:

We agree that the mass transfer is important and could strongly influence the catalytic process. Following the reviewer's suggestion, we measured the porosity of the Co@Si_x catalysts. As shown in Reviewer-Only Table 1, the Co@Si_x catalysts show similar surface areas at 134.6–157.6 m²/g and mesoporous volumes at 0.17–0.22 cm³/g. The results mean comparable mass transfer on these catalysts, which should not be the reason for the distinguishable catalytic performances.

The cobalt–silica interaction was further studied on the Co@Si_x catalysts with different Co/Si ratios. Reviewer-Only Figure 16 shows H₂-TPR profiles of these samples. The Co₃O₄ without silica species showed the reduction peaks at 387 and 494 °C, which are assigned to the reduction of Co³⁺ to Co²⁺ and Co²⁺ to Co⁰. When the cobalt oxide was modified by silica, these peaks were remarkably shifted to higher temperature and the Co²⁺ to Co⁰ signal was weakened. For example, the Co₃O₄@Si_{0.52} showed the Co³⁺ to Co²⁺ signal at 402 °C and weak Co²⁺ to Co⁰ signal at 533 °C. The sample with lower Co/Si ratio, such as the Co₃O₄@Si_{0.95} sample, exhibited even higher Co³⁺ to Co²⁺ signal at 410 °C and almost undetectable Co²⁺ to Co⁰ signal. These data demonstrate the different silica amount strongly influences the oxidation state of cobalt nanoparticles, which is further confirmed by the Co 2p XPS (Figure S24 in the revised SI) and Raman spectra (Figure S31 in the revised SI).

The relationship between the methanol yield in CO₂ hydrogenation and Co⁰/Co²⁺ ratio over various catalysts are presented in Reviewer-Only Figure 23. Compared with CoO_x catalyst, the Co@Si_{0.52} and Co@Si_{0.95} with CoO species exhibited enhanced methanol yields, which is reasonably due to the silica modification effect. Further increasing the silica content led to the decrease of methanol yield over the Co@Si_{1.48} and Co@Si_{1.87} catalysts. These data confirm that the balanced metallic Co and CoO phases on the catalysts are important for the methanol production. More Co⁰ species caused the formation of a large amount of methane with poor methanol selectivity. The

sole CoO phase is known to have poor activity for the hydrogenation. Therefore, the Co@Si_{0.95} catalyst with balanced phases exhibited the best performance among these samples. Apart from the cobalt oxidation state, more silica species might block more surface sites of the Co@Si_x catalysts, which also influences the catalytic performance. We have added the related results and discussion in our revised manuscript.

Minor Comments:

2-6. Comment: *The XRD patterns on Figure S1 should be normalized to show crystalline structure of the catalysts.*

Responses: Thanks for the comments. Yes, we have normalized the XRD patterns to show crystalline structure (Reviewer-Only Figure 24) in the revised manuscript.

2-7. Comment: *The writing has several grammatical errors and should be carefully edited and proofread for clarity.*

Responses: Thanks for the comments. Yes, we have modified the description for clarity and carefully checked the revised manuscript to avoid grammatical mistakes.

2-8. Comment: *The supplemental information should have a table of content to guide the reader to the relevant information, given the length of the document.*

Responses: Thanks for the comments. Yes, in the revised manuscript, we provided a table of content for the guidance and convenience of readers.

2-9. Comment: *Was any surface enrichment of the cobalt species for the Co@Si_x observed for these materials?*

Responses: Thanks for the valuable comments. We detected the dispersion of Co and Si on the different depth of the sample with etching-XPS technique. As shown in Reviewer-Only Figure 25, the atomic ratio of Co and Si on Co@Si_{0.95} catalyst has no obvious change by using carbon signal as a standard, suggesting the uniform dispersion of Co species. We have added these data in the revised manuscript.

2-10. Comments: *The EXAFS radius values reported in Table S3 should be truncated to three significant figures (i.e. 2.05 instead of 2.054) given the accuracy of most XAFS modeling software is at best 0.01 Å.*

Responses: Thanks for the comments. Yes, we have modified the data of Table S4 (Table S3 in the original version) according to the reviewer's suggestion in the revised SI.

Comments from Reviewer #3

This paper reported a novel Co-based catalyst for CO₂ hydrogenation to methanol.

They found the silica acts as powerful support and ligand to modify the cobalt species via a Co–O–SiO_n linkage for turning the cobalt nanoparticles from catalysts for methanation/RWGS into catalysts for methanol production. The obtained results may benefit for further technology development of the production of methanol from CO₂ hydrogenation. In addition, the manuscript is well organized and clearly written. Some revisions are required before final acceptance by Nature Communications, as detailed in the following comments.

3-1. Comments: It was well known that cobalt catalysts have no active sites for WGS or RWGS. However, the CO selectivity over Co/SiO₂ or CoO_x (Traditional cobalt catalyst) is very high. How did the CO form over Co/SiO₂ or CoO_x?

Responses: We are grateful for the valuable comments. Yes, we highly agree that Co-based catalyst is promising for CO₂ methanation, which have been reported previously, but CO is usually formed in these reactions. The methane and CO selectivities depend on the multiple factors (e.g. CO₂ conversion, reaction temperature, pressure, catalyst amount, and the support, *ChemCatChem* **2019**, *11*, 4884–4893; *Catal. Sci. Technol.* **2016**, *6*, 7449–7460; *Catal. Today* **2019**, *337*, 162–170; *ACS Catal.* **2019**, *9*, 2739–2751; *Appl. Catal. B* **2018**, *220*, 397–408; *J. CO₂ Util.* **2018**, *26*, 221–229; *ACS Catal.* **2016**, *6*, 2811–2818; *J. Am. Chem. Soc.* **2017**, *139*, 9739–9754; *Angew. Chem. Int. Ed.* **2014**, *53*, 6705–6709; *Angew. Chem. Int. Ed.* **2016**, *55*, 7968–7973; *Ind. Eng. Chem. Res.* **2013**, *52*, 2247–2256). In our experiments, we also found that the Co/SiO₂ and CoO_x samples with metallic cobalt species exhibited higher methane selectivity than the Co@Si_x catalyst in CO₂ hydrogenation. In these cases, the methane selectivity is still slightly lower than CO (Figure 2a in the main text), which should be due to the employed reaction conditions, and similar phenomena have been observed previously (*ChemCatChem* **2019**, *11*, 4884–4893; *ACS Catal.* **2019**, *9*, 2739–2751; *ACS Catal.* **2016**, *6*, 2811–2818; *J. Am. Chem. Soc.* **2017**, *139*, 9739–9754). On the cobalt catalysts, the CO might from the incomplete hydrogenation of CO₂ (*CO intermediate), decomposition of formate intermediate (*HCOO), and/or dehydrogenation of methoxy intermediate (*CH₃O) (*J. Am. Chem. Soc.* **2017**, *139*, 9739–9754; *ChemCatChem* **2019**, *11*, 4884–4893; *Angew. Chem. Int. Ed.* **2014**, *53*, 6705–6709; *ACS Catal.* **2019**, *9*, 2739–2751; *Catal. Sci. Technol.* **2016**, *6*, 7449–7460; *ACS Catal.* **2016**, *6*, 2811–2818; *Angew. Chem. Int. Ed.* **2013**, *52*, 4406–4409; *J. Phys. Chem. C* **2018**, *122*, 20279–20288; *ChemSusChem* **2015**, *8*, 260–263; *Surf. Sci.* **2005**, *598*, 128–135; *Appl. Catal. B* **2010**, *99*, 257–264; *J. Catal.* **2008**, *256*, 24–36; *J. Phys. Chem. C* **2019**, *123*, 9139–9145). For example, our TPSR-MS investigation reveals that the *CH₃O could be transformed into CO product on the cobalt catalysts (Reviewer-Only Figure 9).

In addition, we explored the methane and CO selectivity as a function of reaction temperature in the Co/SiO₂ and CoO_x catalyzed CO₂ hydrogenation. As shown in Reviewer-Only Figures 5a and 6a, the methane selectivity remarkably increased with the reaction temperatures. For example, the Co/SiO₂ gave CH₄ selectivity at 24.7% at 300 °C. By increasing the reaction temperature to 380 °C, the CH₄ selectivity raised to 54.9%. For the CoO_x catalyst, the CH₄ selectivity was 35.4% at 300 °C and then

increased to 73.2% at 380 °C. Further increasing the reaction temperature to 500 °C gave lower CH₄ selectivity at 52.2% on the Co/SiO₂ catalyst (Reviewer-Only Figures 7a). This can be attributed to the strong exothermic nature of the CO₂ methanation, leading to thermodynamically favorable CO formation *via* reverse water–gas shift reaction and decrease of CH₄ selectivity (*J. CO₂ Util.* **2017**, *17*, 312–319). These data suggest that the reaction temperatures have a significant effect on the CH₄ and CO selectivity (*ChemCatChem* **2019**, *11*, 4884–4893; *Catal. Sci. Technol.* **2016**, *6*, 7449–7460; *ACS Catal.* **2016**, *6*, 2811–2818).

In addition to the reaction temperature, we also explored the influence of reaction pressure, feed gas composition, and catalyst amount to the methane selectivity in CO₂ hydrogenation (Reviewer-Only Figures 7b–d). The higher reaction pressure, more hydrogen feed, and larger catalyst amount (low GHSV) are regarded to be the factors for enhancing the methane selectivity.

We found that the different supports for cobalt species also significantly influenced the CH₄ selectivity in CO₂ hydrogenation. Reviewer-Only Figure 8 shows the data characterizing the performance of Co nanoparticles (20 wt%) supported on SiO₂, TiO₂ (anatase and rutile), Al₂O₃ (denoted as 20Co/SiO₂, 20Co/TiO₂ and 20Co/Al₂O₃, respectively) in CO₂ hydrogenation. The 20Co/Al₂O₃ and 20Co/TiO₂(rutile) exhibited much higher methane selectivity than other catalysts. For example, the 20Co/TiO₂(rutile) gave the highest CH₄ selectivity of 76.3% at 400 °C, while 20Co/Al₂O₃ showed CH₄ selectivity at 58.1%. The silica supported catalysts (20Co/SiO₂ and Co/SiO₂) showed medium CH₄ selectivity at 42.5–45.1%. Notably, the 20Co/TiO₂(anatase) exhibits a really low CH₄ selectivity of 26.5%. This phenomenon is in good agreement with the previous work (*ACS Catal.* **2019**, *9*, 2739–2751; *Angew. Chem. Int. Ed.* **2014**, *53*, 6705–6709; *J. Am. Chem. Soc.* **2017**, *139*, 9739–9754), which might be attributed to the distinguishable interaction between the cobalt and the oxide support.

On the basis of the aforementioned investigation, we confirm that Co-based catalysts are active for CO₂ methanation. Their catalytic performance, for instance, CO₂ conversion and CH₄ selectivity, are strongly influenced by many factors. Therefore, under the given reaction conditions in Figure 2a, the methane selectivity of Co/SiO₂ and CoO_x are reasonable. We have partially added these data and discussion in the revised manuscript.

3-2. Comments: *The CO₂ conversion was very low (~8%) even at very high temperature (320 °C), which is not beneficial for the development of methanol synthesis from CO₂ hydrogenation. How to further enhance the catalytic activity?*

Responses: Thanks for the comments. Yes, the CO₂ conversion was at ~8% for realizing the high methanol selectivity, giving the methanol productivity at 59.7 mmol g_{cat}⁻¹ h⁻¹, which is even higher than the efficient catalysts in literatures (*ACS Catal.* **2015**, *5*, 5827–5836; *Angew. Chem. Int. Ed.* **2016**, *55*, 12708–12712; *J. Am. Chem. Soc.* **2016**, *138*, 12440–12450). In addition, we highly agree that further enhancement of the catalytic activity of Co@Si_x catalysts is important. Inspired by the known high activity

of Cu-based catalysts in CO₂ hydrogenation to methanol, we incorporated Cu species into the cobalt catalysts, the resulted CuCo@Si_{0.88} catalyst exhibited enhanced activity. As shown in Reviewer-Only Figure 26a, the CuCo@Si_{0.88} showed CO₂ conversion of 1.6%–20.5% at 220–340 °C, which are higher than those on the Co@Si_{0.95} catalyst. The CuCo@Si_{0.88} gave methanol as a main product (selectivity of 62.8%–78.8%) at lower reaction temperature (220–280 °C), accompanied with by-products of CO and CH₄. In addition, C₂H₅OH and C₂₊ hydrocarbons are also detected. Compared with the Co@Si_{0.95} catalyst, CuCo@Si_{0.88} exhibits much higher methanol yield at low reaction temperature (33.1 and 46.6 mmol g_{cat}⁻¹ h⁻¹ at 260 and 280 °C, respectively, which are 2.1 and 1.5 times greater than those of Co@Si_{0.95}). These data are even comparable to the performance of cobalt catalyst at high temperature (300–340 °C, Reviewer-Only Figure 26b). On the basis of these primary experimental results, we conclude that the copper incorporation might be a promising method to strengthen the cobalt catalysts.

Because the catalytic study on Cu-incorporated catalyst is only a primary result, and the CuCo structure still need further investigation, these data were not included in the revised manuscript but just shown here for answering the reviewer's comments.

3-3. Comments: *From Fig. 2a, the methanol selectivity over Co/SiO₂ is much higher than that over CoO_x at 320 °C, even is higher than that over Cu-based catalysts. However, the trace *CH₃O signal was found both on Co/SiO₂ and CoO_x. Why? Maybe the temperature is high (350 °C) for In-situ IR measurement.*

Responses: Thanks for the comments. Under the given reaction conditions in Figure 2a, both Co/SiO₂ and CoO_x exhibited very low methanol selectivity at 16.6% and 3.1%, respectively. The methanol product was almost undetectable on Cu/ZnO/Al₂O₃ catalyst because the given reaction conditions are not optimal for Cu-based catalysts (the detailed data on the performance of Cu/ZnO/Al₂O₃ catalyst are shown in Reviewer-Only Figures 13 and 14).

In Figure 2a, the Co/SiO₂ and CoO_x gave CO and CH₄ as dominant products, which is in good agreement with those in the IR data, where the DRIFT spectra showed major peaks at 1268, 1370, 2858, 2960 and 3020 cm⁻¹ for Co/SiO₂, and 1338, 1595, 2890 and 2946 cm⁻¹ for CoO_x with very weak signal of *CH₃O. Such phenomena are due to the rich metallic cobalt on these two catalysts. Considering the very low methanol selectivity on these two catalysts (12.6% vs 2.5% at 340 °C), it is difficult to link the *CH₃O signal intensities in DRIFT spectra to the methanol selectivity in the practical tests. The slightly higher methanol selectivity on Co/SiO₂ than CoO_x might be due to the silica support and/or nanosized cobalt particles.

3-4. Comments: *The CH₄ selectivity is very low over Co@Si_x catalysts. What is the mechanism for this phenomenon, low H₂ dissociation ability or high stability of *CH₃O intermediate species? The discussion on this point is not very clear.*

Responses: Thanks for the comments. Yes, the CH₄ selectivity is very low over the Co@Si_x catalysts, which should be due to the contribution of both low

hydrogenation ability and high $^*\text{CH}_3\text{O}$ stability. The CH_4 was formed in most of the CO_2 hydrogenation reactions, and multiple reaction pathways and mechanisms have been proposed (*Angew. Chem. Int. Ed.* **2016**, *55*, 737–741; *J. Am. Chem. Soc.* **2017**, *139*, 6827–6830; *Chem. Sci.* **2019**, *10*, 3161–3167; *Angew. Chem. Int. Ed.* **2016**, *55*, 7968–7973; *J. Catal.* **2016**, *343*, 115–126; *Appl. Catal. A* **2014**, *470*, 405–411; *J. Am. Chem. Soc.* **2016**, *138*, 6298–6305; *Appl. Surf. Sci.* **2015**, *351*, 504–516; *Catal. Sci. Technol.* **2016**, *6*, 4048–4058; *Phys. Chem. Chem. Phys.* **2013**, *15*, 5701–5706). Among these routes, the deep C–O cleavage is regarded to be the key step for CH_4 formation. In the catalysis over the $\text{Co}@\text{Si}_x$ catalysts, the $^*\text{CH}_3\text{O}$ was abundantly formed and regarded as a crucial intermediate for methanol production. However, the $^*\text{CH}_3\text{O}$ could also be transformed *via* C–O cleavage to form methane on the generally supported cobalt catalyst as confirmed by the TPSR-MS characterization (Reviewer-Only Figure 9). Interestingly, the $\text{Co}@\text{Si}_x$ exhibited undetectable methane signal in the TPSR profile, confirming the stabilized C–O bond in the $^*\text{CH}_3\text{O}$ species, which is unfavorable for the methane formation in the reaction process.

In addition, because the reaction system contains abundant hydrogen, it is generally known that the strong H_2 dissociation ability could accelerate the C–O cleavage and deep hydrogenation (*J. Am. Chem. Soc.* **2019**, *141*, 8482–8488; *Angew. Chem. Int. Ed.* **2019**, *58*, 11242–11247). The H–D exchange experiment reveals that the $\text{Co}@\text{Si}_x$ catalyst has lower activity for H_2 dissociation than that on the generally supported cobalt catalyst (Reviewer-Only Figure 27), which is due to the reduced metallic cobalt sites by the silica modification.

We have added these data and discussion in the revised manuscript.

3-5. Comments: *The author found that the CO_2 conversion was not related to H_2 dissociation ability. Why did $\text{Co}@\text{Si}_{0.95}$ exhibit higher CO_2 conversion? Which is the key factor?*

Responses: Thanks for the comments. Yes, the $\text{Co}@\text{Si}_{0.95}$ catalyst exhibits slightly higher CO_2 conversion than that of traditional Co/SiO_2 or CoO_x catalysts, which is due to the significant advantage of $\text{Co}@\text{Si}_{0.95}$ catalyst in transforming the intermediates rather than activating the reactants.

For a general reaction on the heterogeneous catalysts, the adsorption and activation ability to the substrate could influence the reaction, but the rate is also sensitive to the reaction barrier of the crucial elementary steps in the whole reaction process. The balanced rates for the formation and further transformation of key intermediate is important to obtain high conversion.

The previous DFT simulations have revealed that a moderate dissociation barrier for the substrate benefits the high conversion. Lower reaction barrier benefits the molecular dissociation and form the intermediate to block the catalyst surface, and higher barrier is unfavorable for the molecular dissociation (*ACS Catal.* **2018**, *8*, 10148–10155; *J. Am. Chem. Soc.* **2015**, *137*, 477–482; *J. Am. Chem. Soc.* **2016**, *138*, 6396–6399). Such viewpoint has been confirmed experimentally. For example, in CO hydrogenation, the Rh catalyst is known to have high ability for CO and hydrogen

dissociation, but the supported Rh catalysts usually exhibit low CO conversion (*J. Catal.* **2014**, *313*, 149–158; *ACS Catal.* **2017**, *7*, 5746–5757; *Angew. Chem. Int. Ed.* **2019**, *58*, 8709–8713), because of the easy formation and strong adsorption of *CH₂ intermediates to block the Rh surface. In contrast, the RhMn catalysts, which exhibit weaker adsorption/activation ability to the H₂ and CO molecules, show much higher CO conversion.

On the basis of the aforementioned knowledge, we conclude that the key factor to affect the CO₂ conversion is the moderate dissociation barrier for the reactants and fast transformation of the key intermediates. Possibly, the metallic cobalt easily activates the CO₂ and hydrogen to form abundant intermediates, such as the CH_x, *HCOO, and *CH₃O species, which might block the active sites and delay the reaction process (*J. Am. Chem. Soc.* **2017**, *139*, 9739–9754; *Science* **2017**, *355*, 1296–1299; *J. Am. Chem. Soc.* **2016**, *138*, 12440–12450). In contrast, the Co@Si_{0.95} catalyst with limited accessible sites and moderate activation ability for reactants might benefit the formation and fast transformation of the intermediates, which should contribute the high CO₂ conversion. This result is in good agreement with the phenomena in the encapsulated catalysts (*Nat. Mater.* **2007**, *6*, 507–511; *J. Am. Chem. Soc.* **2015**, *137*, 477–482; *J. Am. Chem. Soc.* **2016**, *138*, 7880–7883; *ACS Catal.* **2012**, *2*, 1958–1966; *J. Am. Chem. Soc.* **2010**, *132*, 8129–8136; *Fuel* **2018**, *215*, 226–231).

We have partially added the discussion in the revised manuscript.

Reviewer-Only Figure & Table

Reviewer-Only Figure 1. *In-situ* Co 2p XPS spectra of $\text{Co}_3\text{O}_4/\text{SiO}_2$ in 0.1 mbar of H_2 .

Reviewer-Only Figure 2. *In-situ* Co 2p XPS spectra of Co@Si_{0.95} catalyst in 0.1 mbar of H₂.

Reviewer-Only Figure 3. (a) XRD pattern and (b) XPS spectrum of Co@Si_{0.95} after high-pressure H₂ reduction (10% H₂ in Ar, 2.0 MPa) at 600 °C for 2 h.

Reviewer-Only Figure 4. (a) XRD pattern and (b) XPS spectrum of Co@Si_{0.95} after CO₂ hydrogenation for 100 h. The reactions are the same as those in Figure 2d in the main text.

Reviewer-Only Figure 5. (a) Dependences of the CO₂ conversion, products selectivity, and (b) carbon balance on temperature over Co/SiO₂ catalyst. Reaction conditions: 0.2 g of catalyst, 2.0 MPa, H₂/CO₂ = 3:1, GHSV = 6000 mL/g h.

Reviewer-Only Figure 6. (a) Dependences of the CO₂ conversion, product selectivity, and (b) carbon balance on temperature over CoO_x catalyst. Reaction conditions: 0.1 g of catalyst, 2.0 MPa, H₂/CO₂ = 3:1, GHSV = 12000 mL/g h.

Reviewer-Only Figure 7. Data characterizing the influences of on (a) temperature, (b) pressure, (c) H₂/CO₂ ratio, and (d) velocity to CO₂ conversion and CH₄ selectivity over Co/SiO₂ catalyst. Standard conditions: 0.2 g of catalyst, 2.0 MPa, H₂/CO₂ = 3:1, 300 °C, GHSV = 6000 mL/g h.

Reviewer-Only Figure 8. Data characterizing the performances of various cobalt catalysts in CO₂ hydrogenation. (a) CO₂ conversion and (b) CH₄ selectivity. Reaction conditions: 0.2 g of catalyst, 0.1 MPa, H₂/CO₂ = 3:1, GHSV = 6000 mL/g h.

Reviewer-Only Figure 9. MeOH-TPSR profiles of (a) Co@Si_{0.95} and (b) Co/SiO₂ catalysts. The CO signal centered at 280 °C characterizing the Co/SiO₂ catalyst. In contrast, no CO signal was observed in equivalent experiments with the Co@Si_{0.95} catalyst at temperature < 300 °C, evidencing the enhanced ability of Co@Si_{0.95} catalyst in anti-dehydrogenation.

Reviewer-Only Figure 10. XRD pattern of used Co/SiO₂ in CO₂ hydrogenation for 100 h. The reaction conditions are the same as those in Figure 2d in the main text.

Reviewer-Only Figure 11. (a–c) HRTEM images and (d) Co nanoparticle size distribution of Co/SiO₂ after CO₂ hydrogenation for 100 h. The reaction conditions are the same as those in Figure 2d in the main text.

Reviewer-Only Figure 12. (a, b) TEM images, (c) Co nanoparticle size distribution, and (d) HRTEM image of Co@Si_{0.95} after CO₂ hydrogenation for 100 h. The reaction conditions are the same as those in Figure 2d in the main text.

Reviewer-Only Figure 13. Dependences of the CO₂ conversion and product selectivity on temperature over Cu/ZnO/Al₂O₃ catalyst. Reaction conditions: 0.2 g of catalyst, 2.0 MPa, H₂/CO₂ = 3:1, GHSV = 6000 mL/g h.

Reviewer-Only Figure 14. Data showing the durability of Cu/ZnO/Al₂O₃ catalyst in CO₂ hydrogenation. Reaction conditions: 0.2 g of catalyst, 2.0 MPa, H₂/CO₂ = 3:1, 240 °C, GHSV = 6000 mL/g h.

Reviewer-Only Figure 15. Comparison of activity of Co@Si_{0.95} and Co@Si_{0.95}-Na catalysts in CO₂ hydrogenation. Reaction conditions: 0.2 g of catalyst, 2.0 MPa, H₂/CO₂=3:1, GHSV= 6000 mL/g h.

Reviewer-Only Figure 16. H₂-TPR profiles of Co₃O₄, Co₃O₄@Si_{0.52}, Co₃O₄@Si_{0.95}, Co₃O₄@Si_{1.48}, and Co₃O₄@Si_{1.87} samples.

Reviewer-Only Figure 17. The kinetic plots of Co@Si_{0.95} and Co/SiO₂ in CO₂ hydrogenation on the basis of (a) H₂ partial pressure and (b) CO₂ partial pressure.

Reviewer-Only Figure 18. XRD patterns of SiO_n before and after H₂ treatment.

Reviewer-Only Figure 19. Co K-edge XANES spectra of Co@Si_x and Co/SiO₂ samples.

Reviewer-Only Figure 20. (a) Co K-edge XANES spectra and (b) first-order derivative spectra of Co₃O₄@Si_x and Co₃O₄/SiO₂ samples.

Reviewer-Only Figure 21. Co 2p XPS spectra of various $\text{Co}_3\text{O}_4@\text{Si}_x$ catalysts.

Reviewer-Only Figure 22. Arrhenius plots for CO, CH₃OH, and CH₄ formation over (a) Co@Si_{0.95} and (b) Co/SiO₂ catalysts.

Reviewer-Only Figure 23. The relationship between methanol yield with $\text{Co}^0/\text{Co}^{2+}$ ratio in CO_2 hydrogenation over various catalysts.

Reviewer-Only Figure 24. XRD patterns of (a) $\text{Co}_3\text{O}_4@Si_x$ and (b) $\text{Co}@Si_x$ catalysts.

Reviewer-Only Figure 25. Etching-XPS analysis of Co@Si_{0.95} catalyst. The etch depth was calculated from the each cycles.

Reviewer-Only Figure 26. (a) Dependences of the CO₂ conversion and product selectivity on temperature in CuCo@Si_{0.88} catalyzed CO₂ hydrogenation reaction. Reaction conditions: 0.2 g of catalyst, 2.0 MPa, H₂/CO₂ = 3:1, GHSV = 6000 mL/g h. (b) Dependences of methanol yield on temperature over CuCo@Si_{0.88} and Co@Si_{0.95} catalysts.

Reviewer-Only Figure 27. Data characterizing the performances of Co/SiO₂ and Co@Si_{0.95} catalysts in H–D exchange test (H₂ + D₂ = 2HD).

Reviewer-Only Scheme 1. Scheme showing the different cobalt sites in Co@Si_x catalysts. For both the cobalt particles on the surface and embedded within the silica sheath, the bare cobalt species are Co⁰ and the cobalt–silica interface has abundant Co^{δ+}, which are highlighted by light blue and red arrows, respectively.

Reviewer-Only Table 1. The surface area and volume data of various samples.

Entry	Sample	S_{BET} (m²/g)	V_{meso} (cm³/g)
1	CoO _x	59.7	0.16
2	Co@Si _{0.52}	151.3	0.22
3	Co@Si _{0.95}	147.4	0.17
4	Co@Si _{1.48}	134.6	0.18
5	Co@Si _{1.87}	157.6	0.20
6	SiO _n	111.4	0.15
7	Co/SiO ₂	123.5	0.26

S_{BET}: BET specific surface area; V_{meso}: mesopore volume.

REVIEWERS' COMMENTS:

Reviewer #1 (Remarks to the Author):

The revision has been made accordingly to my comments and improved significantly. I recommend acceptance of this paper in this form.

Reviewer #2 (Remarks to the Author):

The authors have addressed all comments in a detailed and appropriate fashion.

Reviewer #3 (Remarks to the Author):

Earlier I have reviewed this manuscript and now consider the revised manuscript as is suitable for publication in Nature Communications . The authors have carefully taken care of the referee' comments by providing additional experimental results. I would like to congratulate the authors on this nice piece of work!